# Hepatic p63 regulates steatosis via IKKβ/ER stress

Begoña Porteiro[1,2], Marcos F. Fondevila[1,2], Teresa C. Delgado[3], Cristina Iglesias[1], Monica Imbernon[1,2], Paula Iruzubieta[3], Javier Crespo[3], Amaia Zabala-Letona[4], Johan Fernø[5], Bárbara González-Terán[6], Nuria Matesanz[6], Lourdes Hernández-Cosido[7], Miguel Marcos[8], Sulay Tovar[1,2], Anxo Vidal[1], Julia Sánchez-Ceinos[2,9], Maria M. Malagon[2,9], Celia Pombo[1], Juan Zalvide[1], Arkaitz Carracedo[4,10,11], Xabier Buque[12,13], Carlos Dieguez[1,2], Guadalupe Sabio[6], Miguel López[1,2], Patricia Aspichueta[12,13], María L. Martínez-Chantar[3] & Ruben Nogueiras[1,2]

p53 family members control several metabolic and cellular functions. The p53 ortholog p63 modulates cellular adaptations to stress and has a major role in cell maintenance and proliferation. Here we show that p63 regulates hepatic lipid metabolism. Mice with liver-specific p53 deletion develop steatosis and show increased levels of p63. Down-regulation of p63 attenuates liver steatosis in p53 knockout mice and in diet-induced obese mice, whereas the activation of p63 induces lipid accumulation. Hepatic overexpression of N-terminal transactivation domain TAp63 induces liver steatosis through IKKβ activation and the induction of ER stress, the inhibition of which rescues the liver functions. Expression of TAp63, IKKβ and XBP1s is also increased in livers of obese patients with NAFLD. In cultured human hepatocytes, TAp63 inhibition protects against oleic acid-induced lipid accumulation, whereas TAp63 overexpression promotes lipid storage, an effect reversible by IKKβ silencing. Our findings indicate an unexpected role of the p63/IKKβ/ER stress pathway in lipid metabolism and liver disease.

[1] Department of Physiology, CIMUS, University of Santiago de Compostela-Instituto de Investigación Sanitaria, Santiago de Compostela 15782, Spain. [2] CIBER Fisiopatología de la Obesidad y Nutrición (CIBERobn), 15706, Spain. [3] CIC bioGUNE, Centro de Investigación Biomédica en Red de Enfermedades Hepáticas y Digestivas (CIBERehd), Technology Park of Bizkaia, Derio 48160, Spain. [4] CIC bioGUNE, Centro de Investigación Biomédica en Red de Cáncer (CIBERonc), Technology Park of Bizkaia, Derio 48160, Spain. [5] KG Jebsen Center for Diabetes Research, Department of Clinical Science, University of Bergen, Bergen, Norway. [6] Centro Nacional de Investigaciones Cardiovasculares Carlos III (CNIC), Madrid, Spain. [7] Bariatric Surgery Unit, Department of General Surgery, University Hospital of Salamanca, Department of Surgery, University of Salamanca, Salamanca, Spain. [8] Department of Internal Medicine, University Hospital of Salamanca-IBSAL, Department of Medicine, University of Salamanca, Salamanca, Spain. [9] Department of Cell Biology, Physiology and Immunology, Instituto Maimónides de Investigación Biomédica de Córdoba (IMIBIC)/University of Córdoba/Hospital Universitario Reina Sofía, Córdoba 14004, Spain. [10] IKERBASQUE, Basque foundation for science, Bilbao 48011, Spain. [11] Department of Biochemistry, University of the Basque Country (UPV/EHU), Spain. [12] Department of Physiology, University of the Basque Country (UPV/EHU), Spain. [13] Biocruces Research Institute, Bilbao 48903, Spain. Correspondence and requests for materials should be addressed to R.N. (email: ruben.nogueiras@usc.es).

NAFLD and NASH are two of the most common liver diseases associated with obesity and the metabolic syndrome[1]. Although 70% of obese subjects will develop NAFLD, the fact that some obese patients do not develop hepatic disease, while a few subjects with normal BMI or discrete overweight develop NAFLD and NASH has prompted the search for different genes that may protect or exacerbate the development of the disease in relation to diet.

p63 belongs to a family of transcription factors constituted by p53, p63 and p73 (ref. 2). p63 and p73 are functional homologues of p53, sharing high-sequence and structural similarities. p63 is a complex gene that encodes multiple isoforms that can be simplified in two categories: isoforms with an acidic transactivation domain (TA isoforms); and isoforms that lack this domain (ΔN isoforms). It is generally accepted that TAp63 functions, as a metastasis suppressor[3], whereas ΔNp63 have oncogenic properties[4].

Alterations in metabolism are crucial for tumour progression and tumour cell survival, and thereby it seems logical that the p53 family members may be deeply involved in the control of certain metabolic and cellular dysfunctions[5]. p53 has been shown to inhibit lipid biosynthesis and to induce fatty acid oxidation[5,6]. However, the link between p53 and hepatic lipid metabolism is currently not fully understood, as some reports indicate that p53 is an essential player in the pathogenesis of alcoholic fatty liver disease[7–9] and NAFLD[10–14], whereas others suggest that it attenuates liver steatosis[15,16]. The conclusions of those studies should be interpreted with caution since they have not evaluated the role of p53 through its manipulation specifically in the liver but rather based on gene expression results, the use of pharmacological compounds, mice lacking p53 globally or in vitro assays. In addition, the effects of p53 on lipid metabolism appears to be different in situations of nutritional stress, being tissue- and context-dependent[6]. Furthermore, p53 functions in normal cells are also confusing because of the compensatory role played by other members of the p53 family, such as p63 (ref. 5). Metabolic studies on p63 are scarce and its role in energy balance is not well understood. Whereas heterozygous p63 mice showed weight loss[17], mice lacking TAp63 develop late-onset obesity, glucose intolerance, insulin resistance and steatosis[18].

p63 can act as a sensor in the cellular response to stress by promoting cell fate decisions and regulate several adaptive responses[19,20]. ER also plays a major role in fatty acid and cholesterol metabolism[21] and ER-transmembrane-signalling molecules regulate lipid metabolism[22,23]. Previous studies indicate that ER stress has an important role in the development and progression of NAFLD[24]. Therefore, we felt that some of the effects of p53 could be likely mediated by p63. Thus, in the present paper we aim to explore this possibility as well as to acquire knowledge on the role of hepatic p63 on NAFLD, and the possible involvement of ER stress in the molecular pathways altered after gain- and loss-of-function of liver p63.

Here, we show that reduced levels of endogenous p53 in the whole body or specifically in the liver of mice lead to increased hepatic lipid content independent of diet. The rescue of hepatic p53 in global p53 null mice is sufficient to attenuate the amount of fat in the liver. The action of p53 on lipid homoeostasis in liver is mediated by p63, as down-regulation of p63 in the liver attenuates liver steatosis in DIO and p53 null mice. The activation of TAp63α in the liver increases hepatic fat content and this is mediated by the activation of IKKβ and ER stress. These results found in preclinical models were supported by data obtained in human hepatocytes and liver biopsies from obese NAFLD patients. The emerging role of p63/IKKβ/ER stress in liver

disease may allow the development of reliable and efficient pharmacotherapies in the treatment of NAFLD.

## Results

**p53 null mice have increased hepatic fat content**. p53 null mice showed the expected shorter lifespan and tumour spectrum (Supplementary Fig. 1)[25]. Male p53 null mice on a high fat diet (HFD) for 11 weeks gained significantly less weight and fat mass relative to WT-controls (Supplementary Fig. 1), they exhibited more lipid droplets within hepatocytes, more triglycerides (TG) in the liver, increased serum AST and ALT levels and up-regulation of mRNA expression of PPARγ and SCARB1 and protein levels of FAS, cleaved caspase 3, cleaved caspase 7, and members of the UPR (pIRE/IRE, XBP1s, pPERK and peIF2α/eIF2α), a cellular response to ER stress[26] that is strongly linked to liver steatosis[27] compared with those from their WT littermates independent of the type of diet received (Supplementary Fig. 2). These significant differences in body weight and body composition found in males were not detected in females (Supplementary Fig. 3).

**Down-regulation of p53 causes hepatic steatosis**. Using AAV8-mediated Cre-LoxP recombination in p53 floxed mice (p53 flox/flox), we aimed to examine the role of the liver-specific down-regulation of p53 on hepatic function. One month after the AAVs injection, we found reduced protein levels of p53 in the liver (Fig. 1a), an increased amount of lipid droplets in hepatocytes, hepatic TG levels and serum AST levels, without changes in serum TG (Fig. 1b,c). In agreement with an impaired hepatic status, protein levels of FAS, pJNK/JNK, several markers of ERstress (pIRE/IRE, XBP1s, pPERK and peIF2α/eIF2α) and cleaved caspases 3 and 7 were increased after the inactivation of hepatic p53 (Fig. 1d). To note, higher XBP1s protein levels were also detected following nuclear isolation (Supplementary Fig. 4). We next measured the protein levels and gene expression of different factors that play a key role in lipid metabolism. More precisely, we focused on FAS (the enzyme that catalyses the synthesis of palmitate); several markers of lipid oxidation such as CPT1 (responsible for the transport of long-chain fatty acid-CoA from cytoplasm into mitochondria for oxidation) and ACADM and ACADL (that catalyse the initial step in each cycle of fatty acid β-oxidation in the mitochondria of cells), and a marker of long-chain fatty acid uptake named FATP2. All these genes remained unchanged in mice with reduced activity of p53 in the liver (Fig. 1e). In addition, serum ketone bodies remained also unaltered (Fig. 1f). Therefore, these findings suggest that knocking-down p53 in the liver stimulates the expression of lipogenic markers as well as members of the UPR, whereas markers for lipid oxidation or lipid uptake were unaffected. Finally, glucose tolerance and insulin sensitivity remained unaltered in these mice fed a standard diet (Fig. 1g,h).

**Hepatic overexpression of p53 ameliorates hepatic steatosis**. We next tested if the recovery of p53 in the liver was sufficient to reverse the hepatic effects of global p53 deficiency. GFP was specifically detected in the liver after the tail vein injection of adenoviruses encoding either GFP or GFP together with p53 (Fig. 2a). Hepatic levels of p53 mRNA were significantly elevated following the injection of Ad-p53 for 1 week compared with mice injected with Ad-GFP (Fig. 2b). After the injection of Ad-p53, both WT and p53 null mice fed a HFD exhibited less lipid droplets in their hepatocytes, decreased hepatic TG and a non-significant tendency to lower levels of serum AST (Fig. 2c,d). messenger RNA (mRNA) expression of ACADM and FATP2 was diminished in WT and p53 null mice following Ad-p53 injection and no changes were detected in CPT1, ACADL (Fig. 2e). The ameliorated hepatic

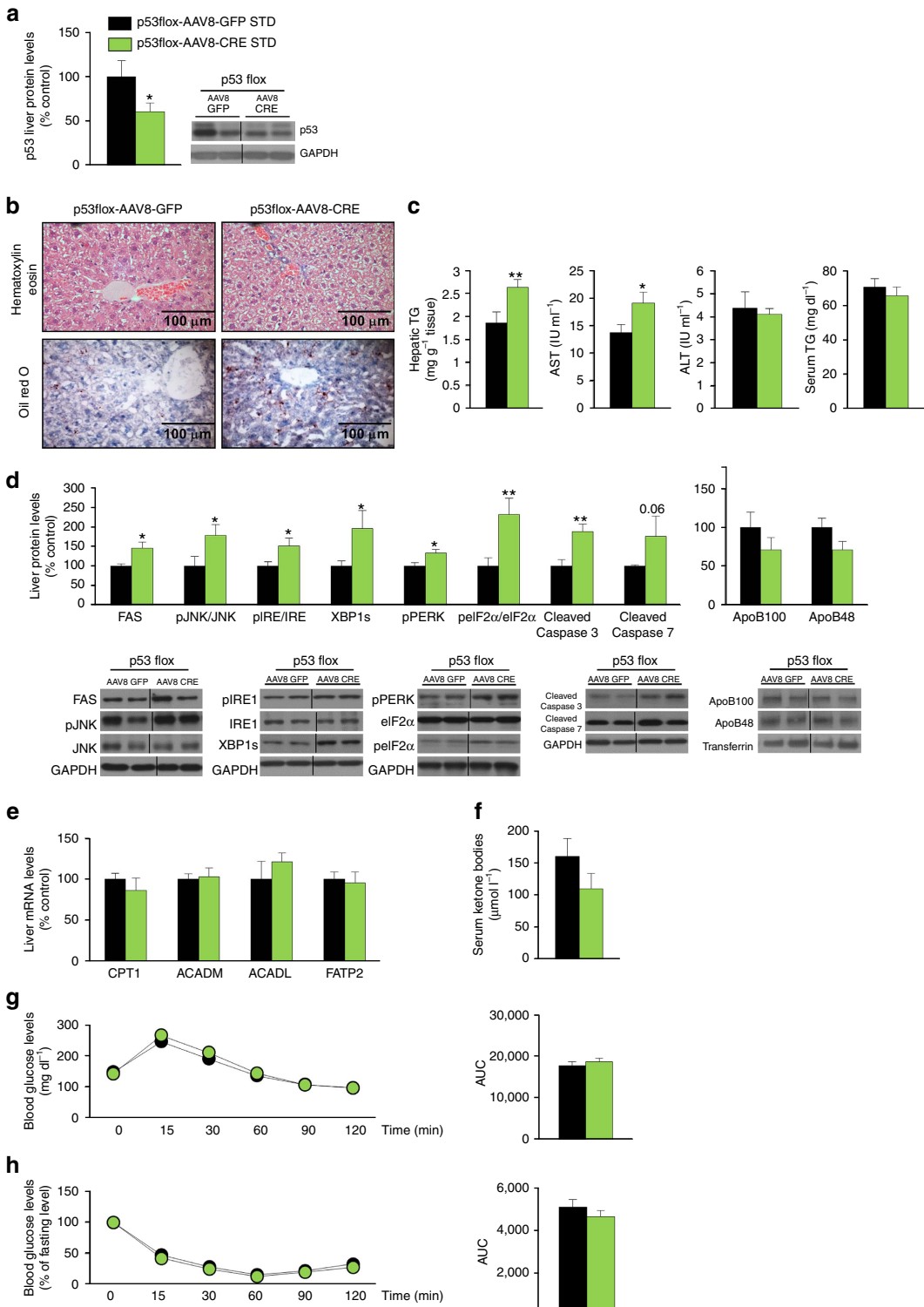

**Figure 1 | Effect of hepatic p53 down-regulation on liver steatosis.** (**a**) p53 protein levels in the liver after a tail vein injection of an associate adenovirus serotype 8 (AAV8) expressing either GFP or Cre in p53floxed mice fed a standard diet (STD) ($n = 7$ per group). Effects of the liver-specific silencing of p53 on (**b**) in hematoxylin-eosin (upper panel) and oil red O staining (lower panel) of mice liver sections ($n = 3$ per group); (**c**) total liver TG content, serum AST, ALT and TG levels ($n = 8$ AAV8-GFP and 10 AAV8-Cre mice); (**d**) liver protein levels of FAS, pJNK/JNK, pIRE/IRE, XBP1s, pPERK, peIF2α/eIF2α, cleaved caspase 3, cleaved caspase 7, ApoB100 and ApoB48; (**e**) mRNA expression of CPT1, ACADM, ACADL and FATP2; (**f**) serum ketone bodies; (**g**) glucose and (**h**) insulin tolerance test ($n = 8$ AAV8-GFP and 10 AAV8-Cre mice). The values of AAV8 GFP mice were always normalized to 100% ($n = 7$ per group). Protein GAPDH or transferrin levels were used to normalize protein levels. Dividing lines indicate splicings in the same gel. Uncropped blots of this Figure accompanied by the location of molecular weight markers are shown in Supplementary Fig. 10. Data are presented as mean ± standard error mean (s.e.m.). Statistical significance, *$P < 0.05$ and **$P < 0.01$, was tested using Student $t$-test comparing AAV8-GFP and AAV8-Cre mice.

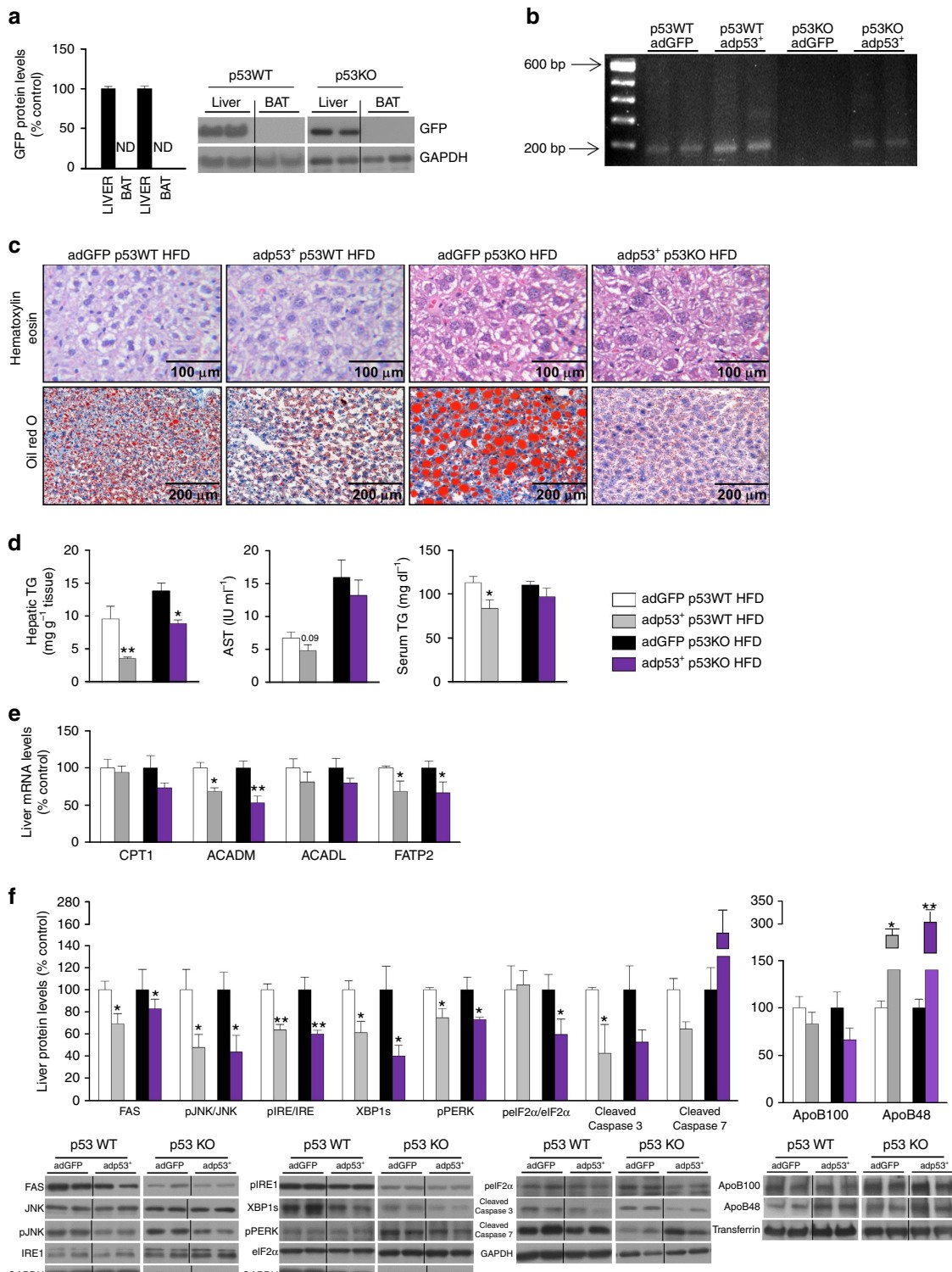

**Figure 2 | Hepatic rescue of p53 in mice fed a high fat diet ameliorates steatosis of global p53 null mice.** (**a**) GFP protein levels in the liver and brown adipose tissue (BAT) of WT and p53 null mice after a tail vein injection of adenoviruses encoding either GFP or p53 ($n = 5$ per group). (**b**) p53 gene expression in the liver of WT and p53 null mice after the injection of adenoviruses encoding either GFP or p53. (**c**) Representative photomicrographs of hematoxylin-eosin (upper panel) and oil red O staining (lower panel) of mice liver sections ($n = 3$ per group); (**d**) total liver TG content, serum AST and TG levels ($n = 6$ GFP and 5 p53 in WT mice; $n = 5$ GFP and 4 p53 in KO mice); (**e**) mRNA expression of CPT1, ACADM, ACADL and FATP2; and (**f**) protein levels of FAS, pJNK/JNK, pIRE/IRE, XBP1s, pPERK, peIF2α/eIF2α, cleaved caspase 3, cleaved caspase 7, ApoB100 and ApoB48 in the liver of WT and p53 null mice fed a HFD after the over-expression of hepatic p53. Western blots were performed separately in WT and p53 null mice, and the values of Ad GFP mice were always set to 100% ($n = 5$ GFP and p53 in both WT and p53 null mice). GAPDH or transferrin were used to normalize protein levels. Dividing lines indicate splicings in the same gel. Uncropped blots of this Figure accompanied by the location of molecular weight markers are shown in Supplementary Fig. 11. Data are presented as mean ± standard error mean (s.e.m.). Statistical significance, *$P < 0.05$ and **$P < 0.01$, was tested using Student $t$-test comparing WT and KO mice. ND: non detected.

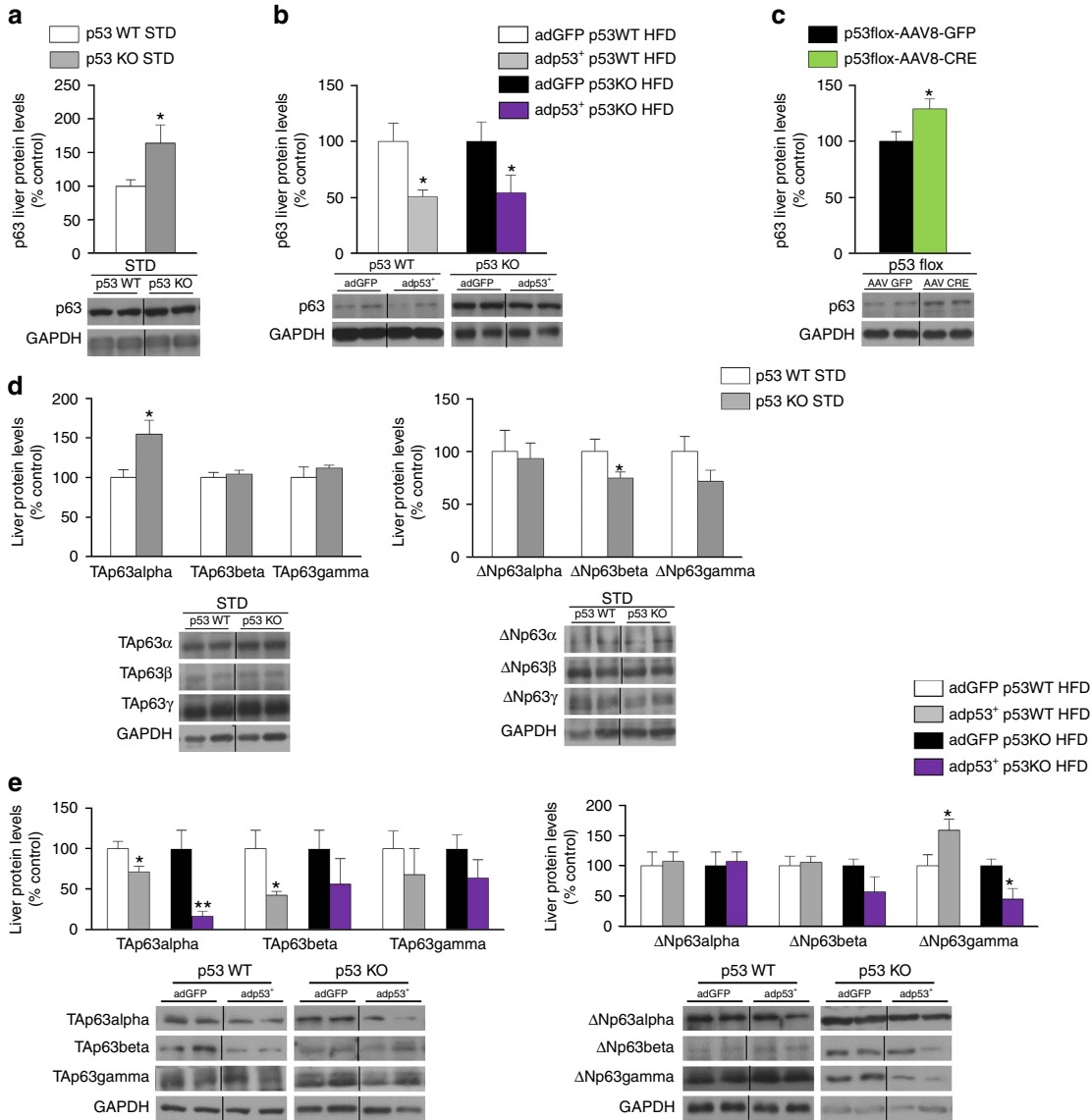

**Figure 3 | p53 modulates p63 in the liver.** (**a**) p63 protein levels in WT and p53 null mice fed a standard diet ($n = 7$ per group). (**b**) p63 liver protein levels in WT and p53 null mice fed a high fat diet injected with p53 dominant positive adenovirus and GFP ($n = 5$ GFP and 4 p53 in both WT and p53 null mice). (**c**) p63 liver protein levels in p53 floxed mice injected with either AAV8-GFP or AAV8-CRE ($n = 7$ per group). (**d**) Protein levels of TAp63 and ΔNp63 isoforms in the liver of WT and p53 null mice fed a standard diet. (**e**) Protein levels of TAp63 and ΔNp63 isoforms in the liver of WT and p53 null mice fed a high fat diet injected with p53 dominant positive adenovirus and GFP. GAPDH was used to normalize protein levels. Dividing lines indicate splicings in the same gel. Uncropped blots of this Figure accompanied by the location of molecular weight markers are shown in Supplementary Fig. 12. Data are presented as mean ± standard error mean (s.e.m.). Statistical significance, *$P < 0.05$ and **$P < 0.01$, was tested using two-tailed Student $t$-test.

fat content in WT and p53 null mice following Ad-p53 injection was associated with the inhibition of hepatic FAS, pJNK/JNK, ER stress markers and cleaved caspase 3 (Fig. 2f).

Several downstream target genes of p53 have previously been linked to the regulation of lipid metabolism in the liver such as p73, bax, p66shc and p21 (refs 13,28) but we failed to detect significant changes in the expression of those genes (Supplementary Fig. 5). Therefore, we assessed the levels of other p53 family members, which have been reported to mediate p53-dependent apoptosis[29]. p73 expression was not changed in the liver of mice lacking p53 (Supplementary Fig. 4). However, hepatic p63 levels were increased in p53 null mice and in mice with specific down-regulation of hepatic p53 (Fig. 3a–c), while they were decreased when p53 expression was recovered in the liver of p53 null mice (Fig. 3b).

**TAp63α is regulated by hepatic p53 and by high fat diet**. p63 is expressed as a full-length protein that retains a full transactivation domain (TAp63) or a N-terminally truncated isoform (ΔNp63) that lacks part of this domain[5]. To investigate the specific isoforms of p63 regulated in our animal models, we next measured the expression of TAp63 and ΔNp63 isoforms in the liver of WT and p53-null mice fed a standard diet. The results indicated that TAp63α is increased while TAp63β and TAp63γ remain unchanged (Fig. 3d). In this animal model, the levels of ΔNp63β and ΔNp63γ decrease, while ΔNp63α does not change (Fig. 3d). Next, we measured the expression of TAp63 and ΔNp63 isoforms in the liver of WT and p53-null mice fed a HFD with or without the exogenous p53 expression in the liver. Our findings show that TAp63α is decreased in both WT and p53-null mice receiving adp53+ injections (Fig. 3e), thereby reproducing the

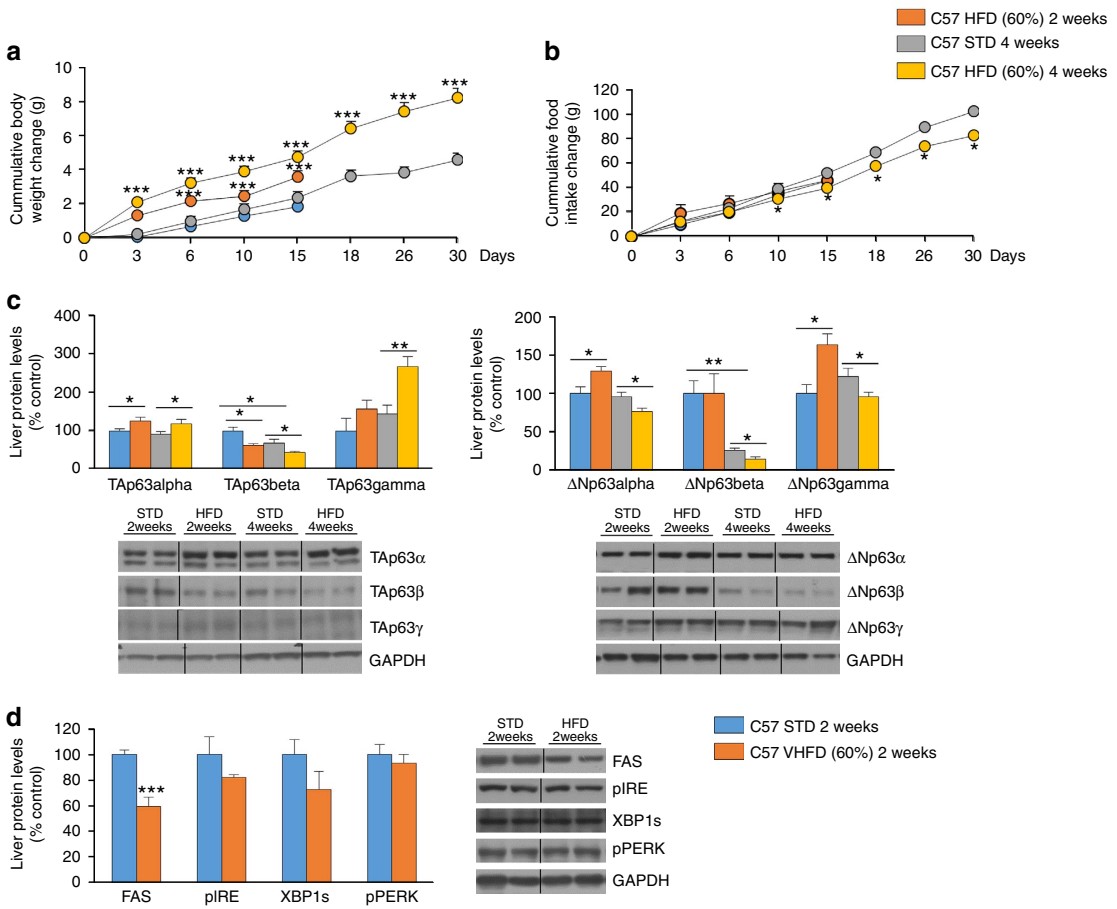

**Figure 4 | High fat diet increases hepatic TAp63α.** (**a**) Body weight; (**b**) cumulative food intake ($n = 10$ per group); (**c**) protein levels of TAp63 and ΔNp63 isoforms; and (**d**) protein levels of FAS, pIRE, XBP1s and pPERK in the liver of mice fed a high fat diet during 2 and 4 weeks ($n = 6$ per group). GAPDH was used to normalize protein levels. Dividing lines indicate splicings in the same gel. Uncropped blots of this Figure accompanied by the location of molecular weight markers are shown in Supplementary Fig. 13. Data are presented as mean ± standard error mean (s.e.m.). Statistical significance, *$P < 0.05$, **$P < 0.01$ and ***$P < 0.001$. For multiple comparison (**a**–**c**) a one way ANOVA followed by Bonferroni or Kruskal-Wallis test was performed. Student $t$-test was used in western blot (**d**) comparing STD and VHFD 2 weeks.

results found in total p63 levels (Fig. 3b). TAp63β was decreased in WT mice treated with adp53+ but not in p53-null mice treated with adp53+; and TAp63γ remained unchanged in both conditions. In this animal model, the levels of ΔNp63α and ΔNp63β did not change, and ΔNp63γ increased in WT mice treated with adp53+ but decreased in p53-null mice treated with adp53+ (Fig. 3e).

Since it is known that diet-induced obesity causes liver steatosis, we fed C57/B6 mice a standard diet (STD) and HFD for 2 or 4 weeks, and then measured by western blot TAp63 and ΔNp63 isoforms in the liver. Mice showed the characteristic increase in body weight (Fig. 4a) and decreased amount of food intake (Fig. 4b). We found that TAp63α was the only isoform with sustained high expression after both 2 and 4 weeks of HFD, whereas TAp63β is decreased and TAp63γ is increased only after 4 weeks of HFD (Fig. 4c). When measuring ΔNp63 isoforms, the results were not consistent as we detected higher levels of ΔNp63α and ΔNp63γ after 2 weeks of HFD, whereas the levels of the three ΔNp63 isoforms decreased only after 4 weeks of HFD (Fig. 4c). Moreover, these HFD-induced changes in TAp63α temporally precede changes in protein levels of FAS and ER stress markers, since FAS levels were reduced while pIRE, XBP1s and pPERK remained unchanged after 2 weeks of HFD (Fig. 4d).

**Inhibition of p63 reduces p53 knockdown-induced steatosis.** Given the effects of p53 manipulation on the expression of p63 isoforms, we next pursued to investigate if the down-regulation of hepatic p63 could reverse the hepatic damage of mice lacking p53 in the liver. To address this question, we used AAV-mediated Cre-LoxP recombination in p53 floxed mice (p53 flox/flox) together with lentivirus expressing GFP alone or a lentivirus encoding a p63 shRNA administered in the tail vein to inhibit expression of both p53 and p63 specifically in liver. Infection efficiency of shRNA-p63, which inhibits total p63, was confirmed by significantly reduced p63 protein levels (Fig. 5a). One month after the injection of p63 shRNA, mice exhibited less lipid droplets in their hepatocytes compared to mice injected AAV8-Cre and GFP lentiviruses (Fig. 5b). We also found a significant decrease in total hepatic TG content, as well as in serum AST and ALT in mice expressing low levels of both p53 and p63 in the liver in contrast to mice lacking hepatic p53 (Fig. 5c). The diminished hepatic steatosis in mice expressing low levels of both p53 and p63 in the liver was also consistent with the reduced levels of hepatic FAS, pJNK and ER stress markers such as pPERK and XBP1s (Fig. 5d), while no changes were found in protein levels of ApoB (Fig. 5d) or mRNA expression of CPT1, ACADM, ACADL and FATP2 (Fig. 5e).

**Inhibition of p63 ameliorates high fat diet-induced steatosis.** We next assessed the efficiency of the hepatic silencing of p63 in DIO mice, another model of hepatic steatosis. One month after the injection of p63-directed shRNA lentivirus, western blot analysis showed significantly reduced hepatic p63 protein levels

(Fig. 5f). Mice exhibited less lipid droplets in their hepatocytes compared with those observed in mice injected with scramble lentiviruses (Fig. 5g). Hepatic TG content, serum AST and serum ALT were also significantly lower in mice where hepatic p63 was silenced (Fig. 5h). The amelioration of hepatic steatosis in DIO

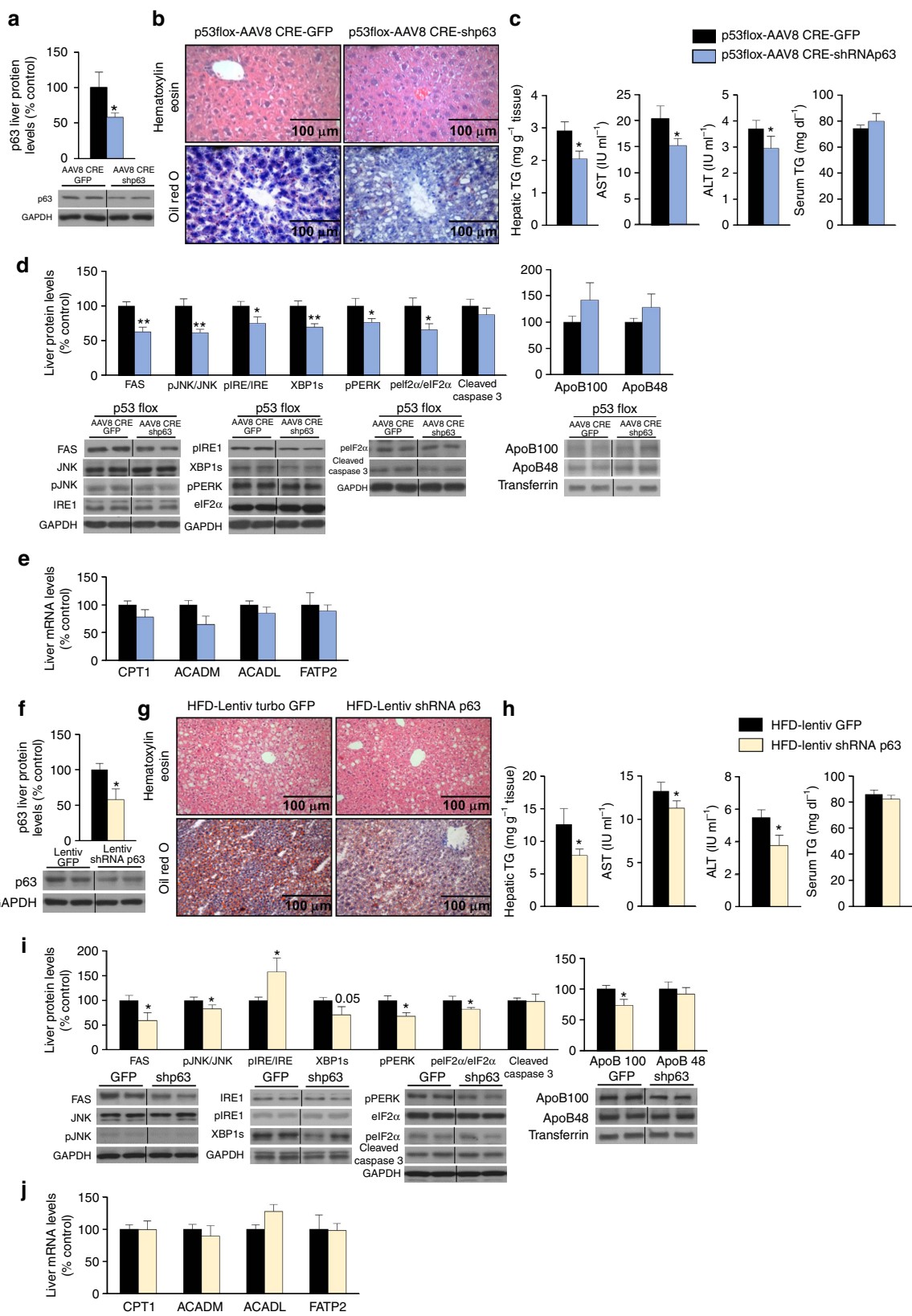

mice expressing low levels of p63 in the liver was also consistent with the inhibition of hepatic FAS, pJNK and some of the ER stress markers such as pPERK and peIF2α/ eIF2α (Fig. 5i). ApoB 100 but not ApoB 48 was reduced after knocking down of p63 in the liver (Fig. 5i). However, no changes were detected in the mRNA expression of CPT1, ACADM, ACADL or FATP2 (Fig. 5j).

**TAp63α overexpression induces steatosis via ER stress**. We next performed *in vivo* adenovirus-associated gene transfer to over-express TAp63α in the liver by tail vein injection of AAV8 encoding either GFP or p63 and treated the mice with tauroursodeoxycholic acid (TUDCA), a chemical chaperone that reduces ER stress[30]. Hepatic protein levels of p63 were significantly elevated following the injection of AAV8-TAp63α for 1 month compared with mice injected with AAV8-GFP (Fig. 6a). Mice with overexpression of p63 in the liver showed more lipid droplets within hepatocytes (Fig. 6b), increased total hepatic TG content and higher levels of serum AST (Fig. 6c). The hepatic fat content following AAV8-TAp63α injection was associated with the stimulation of hepatic ER stress as demonstrated by the up-regulation of several ER stress markers such as XBP1s and pEIF2α/eIF2α (Fig. 6d). Consistent, higher XBP1s protein levels were also detected following nuclear isolation (Supplementary Fig. 6B). All these effects were reduced after the administration of TUDCA during 1 week (Fig. 6b–d). The overexpression of TAp63α in the liver reduced CPT1 and ACADM gene expression but TUDCA did not modify these profiles (Fig. 6e).

In a second approach, we tested the physiological relevance of UPR on the hepatic actions of TAp63α in mice fed a chow diet. For this, we targeted glucose regulated protein 78 (GRP78), a chaperone that facilitates the proper protein folding acting upstream of the UPR[31,32]. Thus, adenoviruses encoding GRP78 wild-type (GRP78 WT) or control adenovirus expressing GFP alone[33] were injected in the tail vein of mice alongside with or without TAp63α (ref. 34). Infection efficiency in the liver was corroborated by western blot of GRP78 (Fig. 6f). The higher amount of lipid droplets and total hepatic TG content, higher levels of serum AST and stimulated ER stress and FAS observed after the overexpression of TAp63α in the liver were all reduced in mice injected with adenoviruses encoding GRP78, while no changes were detected in serum ketone bodies (Fig. 6g–i), or mRNA expression of CPT1, ACADM, ACADL (Fig. 6j).

**IKKβ links p63 to ER stress**. Some reports have shown that ER stress increased lipogenesis, whereas others have indicated that ER stress can also be a consequence of stimulated lipogenesis. To precisely investigate the signalling pathway mediating the effects of p63 on hepatic lipogenesis, we over-expressed TAp63α in the liver for 2 weeks only. We found a significant increase in the TG content (Fig. 7a,b) and protein levels of XBP1s in the liver of

mice over-expressing TAp63α, while no changes were detected in FAS (Fig. 7b), indicating that changes in hepatic ER stress precedes the stimulation of FAS protein levels. This result was confirmed by another study where mice over-expressing TAp63α for 1 month and then treated with the specific FAS inhibitor C75. As expected, C75 reduced FAS activity (Fig. 7e). Moreover, it was able to decrease the TAp63α-induced hepatic TG content (Fig. 7c,d) but did not affect XBP1s (Fig. 7f).

Next, we assessed the mechanism linking p63 and ER stress. Previous reports indicated that p63 acts upstream of IKK[35,36]. Since IKKα (ref. 37) and IKKβ (ref. 38) are involved in lipogenesis, we measured both IKKα and IKKβ and their phosphorylated levels in the liver of mice models after the manipulation of hepatic p63. We found that pIKKβ and IKKβ mRNA expression, but not total IKKβ protein levels, were significantly increased 1 month (Fig. 7g) after the hepatic overexpression of TAp63α, while pIKKβ, but not total protein levels or mRNA levels of IKKβ, were reduced after the inhibition of hepatic p63 in DIO mice (Fig. 7h). However, IKKα and pIKKα levels were unchanged after the hepatic manipulation of p63 (Fig. 7g,h).

**TAp63α regulates lipid content via IKKβ in THLE-2 cells**. We next performed gain- and loss-of-function experiments in THLE-2 cells. First, we over-expressed TAp63α and ΔNp63α and found that transfection with TAp63α but not ΔNp63α increased the amount of lipids in comparison to controls (Fig. 8a), as well as increased protein levels of XBP1s and FAS (Fig. 8b). In addition to the effects on lipid accumulation, ER stress markers and FAS, we also found that TAp63α induces ER defragmentation (Fig. 8c). Second, scramble or siRNAs directed against TAp63α were transfected into THLE-2 cells and challenged with oleic acid. Lipid accumulation was significantly reduced by siRNA-mediated knockdown of TAp63α (Fig. 8d) and accordingly, protein levels of TAp63α, pIKKβ, XBP1s and FAS were also reduced in both groups treated or not with oleic acid (Fig. 8e). Third, we tested the hypothesis that IKKβ mediated the effects of TAp63α on lipid metabolism. Thus, we initially checked that siRNAs directed against IKKβ and transfected into THLE-2 cells might affect lipid metabolism under basal conditions. Under these circumstances, siRNA IKKβ did not alter lipid accumulation, protein levels of XBP1s or FAS (Supplementary Fig. 7). However, when cells were co-transfected with TAp63α and siRNA IKKβ, we found amelioration in all TAp63α-induced effects (Fig. 8f,g).

Finally, to corroborate the specificity of the antibodies used in this study, we have performed two additional studies. First, we have used a control model to detect an increase in the levels of ER stress markers inducing liver steatosis by a methionine-choline-deficient diet, since it is well established that the UPR is activated in models of obesity and steatosis/steatohepatitis[39], and another group of mice with this diet and injected with a viral vector overexpressing GRP78 specifically in the liver[33]. In this

**Figure 5 | Down-regulation of p63 ameliorates p53 knockdown- and high fat diet-induced steatosis.** (**a**) p63 protein levels in the liver of mice with down-regulated p53 fed a standard diet (STD) after a tail vein injection of a lentiviral particle that encodes either GFP or a shRNA p63 ($n = 7$ per group). Effect of the simultaneous liver silence of p53 and p63 on: (**b**) hematoxylin-eosin (upper panel) and oil red O staining (lower panel) of mice liver sections ($n = 3$ per group); (**c**) total liver TG content, serum AST, ALT and TG levels ($n = 8$ per group); (**d**) liver protein levels of FAS, pJNK/JNK, pIRE/IRE, XBP1s, pPERK, peIF2α/eIF2α, cleaved caspase 3, ApoB100 and ApoB48 ($n = 5$ per group); and (**e**) mRNA expression of CPT1, ACADM, ACADL and FATP2. (**f**) p63 protein levels in the liver of mice fed a HFD injected with lentiviruses encoding shRNA p63; (**g**) effect of the hepatic down-regulation of p63 in mice fed a HFD in the hematoxylin-eosin (upper panel) and oil red O staining (lower panel) of mice liver sections ($n = 4$ per group); (**h**) total liver TG content, serum AST, ALT and TG levels ($n = 11$ GFP and 13 sh-RNA p63 mice); (**i**) liver protein levels of FAS, pJNK/JNK, pIRE/IRE, XBP1s, pPERK, peIF2α/eIF2α, cleaved caspase 3, ApoB100 and ApoB48; and (**j**) mRNA expression of CPT1, ACADM, ACADL and FATP2. Protein GAPDH or transferrin levels were used to normalize protein levels, and control values (AAV8 GFP and lentivirus Turbo GFP) were normalized to 100%. Dividing lines indicate splicings in the same gel ($n = 7$ per group). Uncropped blots of this Figure accompanied by the location of molecular weight markers are shown in Supplementary Fig. 14. Data are presented as mean ± standard error mean (s.e.m.). Statistical significance, $*P < 0.05$ and $**P < 0.01$, was tested using Student $t$-test.

experiment we found that MCD-induced levels of ER stress markers are ameliorated after the administration of a viral vector overexpressing GRP78 in the liver (Supplementary Fig. 8). On the other hand, to proof the specificity of the antibody for other isoforms different to TAp63α, we cotransfected cells with TAp63γ

and indeed found that its protein concentration was increased compared to control cells (Supplementary Fig. 9).

**TAp63α increases *de novo* lipogenesis in THLE-2 cells.** We next measured *de novo* lipogenesis, lipid oxidation and lipid turnover

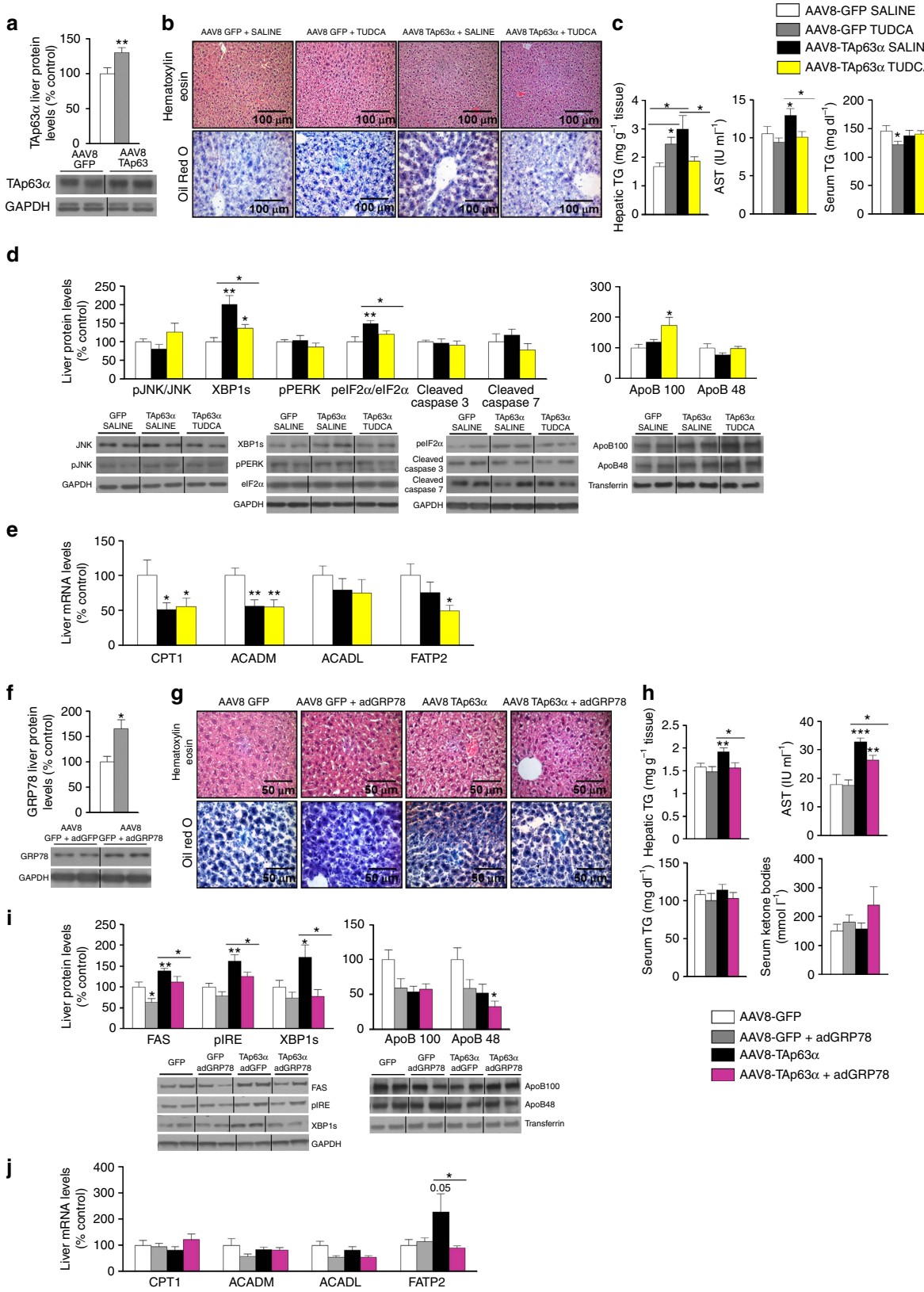

in THLE2 cells transfected with TAp63α. We found that TAp63α significantly increased *de novo* TG and fatty acid synthesis, while *de novo* diacylglycerol lipogenesis showed a non-significant trend to be increased, with no changes in *de novo* fatty acid, phospholipid, esterified cholesterol and free cholesterol biosynthesis (Fig. 9a). Lipid oxidation, reflected as the measurement of palmitic acid oxidation (complete oxidation, left; incomplete oxidation, right) did not show any difference between control and TAp63α cells (Fig. 9b). Finally, the turnover of lipid stores, measured by pulse-chase studies using [³H-oleate] and [¹⁴C-glycerol], did not detect any difference in cellular [³H]-TG, [¹⁴C]-TG, medium [³H]-TG, [¹⁴C]-TG and secreted [³H]-TG, [¹⁴C]-TG (Fig. 9c).

**TAp63, IKKβ and XBP1s in human NAFLD and NASH.**
Analysis of liver biopsies from obese patients with NAFLD (body mass index > 35 kg m⁻²) were obtained from two different cohorts of patients, one used for TAp63 mRNA expression (characterization of patients shown in Supplementary Table 3) and another one used for TAp63 immunohistochemistry (characterization of patients shown in Supplementary Table 4). Our results revealed a positive correlation between p63 and BMI (Fig. 10a), between NAS score and BMI (Fig. 10b) and a clear tendency between p63 and NAS score (Fig. 10c). Moreover, we found elevated mRNA expression of p63, IKKβ and XBP1s compared with individuals without NAFLD and BMI < 35 kg m⁻² (Fig. 10d). Immunohistochemical analysis corroborated that TAp63α is increased in NAFLD and also in NASH patients (characterization of patients shown in Supplementary Table 4) in comparison to lean patients (Fig. 10e). We found that the average NAFLD activity score (NAS), encompassing the degree of steatosis, lobular inflammation and ballooning, is significantly higher in the TAp63α positive samples (we have assigned positive samples as having at least $n = 5$ nuclei of hepatocytes stained for TAp63α) against the negative TAp63α liver samples ($3.6 \pm 0.52$ versus $2.19 \pm 0.28$, $P < 0.05$ Student $t$-test) (Supplementary Table 5).

## Discussion

The relevant role of p53 and p63 in cancer is well established[40–42]. p53 null mice are viable but this protein is implicated in a large number of biological functions, including longevity, stress and ageing[43,44]. In contrast to p53, mice with genetic deletion of p63 have profound developmental alterations leading to early death[45]. In adult cells, the two major functions of p63 are considered to be tumour suppression and cell maintenance and renewal[42]. This latter function of p63 is quite attractive in tissues such as the liver with a high regenerative capacity following injury.

Over the last years, there are accumulating data indicating that p53 is an essential player in liver metabolism, but the results are controversial[7–13,15,16]. Given the current ambiguous literature and that p53 is virtually expressed in all cells and metabolically active tissues, we carried out liver-specific gain- and loss-of-function studies of p53 in order to avoid possible contributions/compensations from p53 modulations in other tissues. We found increased liver fat content in p53 null mice despite the fact that they gain less weight than their WT counterparts when challenged with HFD. A similar exacerbated liver fat content was found in mice with the down-regulation of p53 confined to the liver. The molecular underpinnings triggering lipid storage in the liver seem to include the activation of ER stress and pathways favouring lipid storage. More specifically, our data suggest the presence of increased hepatic *de novo* lipogenesis, with higher levels of FAS in the liver of p53 null mice or after silencing hepatic p53, suggesting an increased hepatic *de novo* lipogenesis. Regarding ER stress, an event that characterizes liver of obese rodents with the metabolic syndrome[46], we found up-regulated protein levels of UPR markers in the liver of p53 null mice, as well as in mice with the liver-specific down-regulation of p53. A specific role of p53 was corroborated by the fact that the rescue of p53 in the liver of p53 null mice attenuated HFD-induced hepatic steatosis by decreasing fatty acid accumulation and ER stress. In the same direction to our present results, previous studies have demonstrated that reduction of p53 at the whole body level caused hepatosteatosis under fasting conditions[15], but the specific tissues involved in this effect was not identified. Taken together, these findings suggest that p53 exerts a profound influence on hepatic metabolism, with loss of p53 leading to harmful effects in the liver. Indeed, our results are clearly independent of nutritional status, as our mice were fed *ad libitum*, but both studies suggest that the lack of p53 impairs hepatic lipid homoeostasis. Therefore we decided to further assess the involvement of p53 downstream signalling pathway in order to unmask potential drug targets.

Hepatic levels of downstream target genes of p53 such as bax or p66shc, which were reported to regulate lipid metabolism[13,28], remained unaltered in our experiments. However, hepatic p63 protein levels were elevated when p53 was lacking or down-regulated, and, more importantly, down-regulation of p63 in the liver attenuated the liver ER stress and fatty acid deposition associated with reduced p53 levels. The interaction between p63 and p53 is strong and they can complement/antagonize each other's functions at the cellular level (for extensive review[41]). The hepatic silencing of p63 was also sufficient to ameliorate diet-induced steatosis, ER stress and fatty acid deposition. In agreement with the results from loss-of-function experiments, overexpression of p63 in the liver increased the levels of ER stress markers, stimulated fatty acid deposition and ultimately induced steatosis, confirming that p63 is a mediator of the effects associated with loss of p53 function. Different reports have shown

**Figure 6 | Hepatic over-expression of TAp63α causes steatosis via ER stress.** (**a**) p63 protein levels in the liver of mice after a tail vein injection of an AAV8 over-expressing either GFP or TAp63α isoform. (**b**) Representative photomicrographs of hematoxylin-eosin (upper panel) and oil red O staining (lower panel) of mice liver sections ($n = 4$ per group); (**c**) total liver TG content, serum AST, ALT and TG levels ($n = 9$ AAV8-GFP and 10 AAV8-TAp63α); (**d**) protein levels of FAS, pJNK/JNK, pIRE/IRE, XBP1s, pPERK, peIf2α/eIF2α, cleaved caspase 3, cleaved caspase 7, ApoB100 and ApoB48; and (**e**) mRNA expression of CPT1, ACADM, ACADL and FATP2 in the liver of mice after hepatic over-expression of TAp63 and IP TUDCA administration ($n = 7$ per group). (**f**) GRP78 protein levels in the liver of mice after a tail vein injection of an Ad over-expressing either GFP or GRP78 ($n = 6$ per group). (**g**) Representative photomicrographs of hematoxylin-eosin (upper panel) and oil red O staining (lower panel) of mice liver sections ($n = 4$ per group); (**h**) total liver TG content, serum AST, TG and ketone bodies levels ($n = 9$ AAV8-GFP and 10 AAV8-TAp63α); (**i**) protein levels of FAS, pIRE/IRE, XBP1s, ApoB100 and ApoB48 ($n = 6$ per group); and (**j**) mRNA expression of CPT1, ACADM, ACADL and FATP2 in the liver of mice after hepatic over-expression of TAp63α and Ad GRP78 administration ($n = 8$ per group). Protein GAPDH or transferrin levels were used to normalize protein levels and control values (AAV8 GFP) were normalized to 100%. Dividing lines indicate splicings in the same gel ($n = 7$ per group). Uncropped blots of this Figure accompanied by the location of molecular weight markers are shown in Supplementary Fig. 15 and Supplementary Fig. 16. Data are presented as mean ± standard error mean (s.e.m.). Statistical significance, $*P < 0.05$, $**P < 0.01$ and $***P < 0.001$. For multiple comparison (**c–e,h–j**) a one way ANOVA followed by Bonferroni or Kruskal-Wallis test was performed. Student $t$-test was used in TAp63alpha and GRP78 liver protein levels (**a,f**).

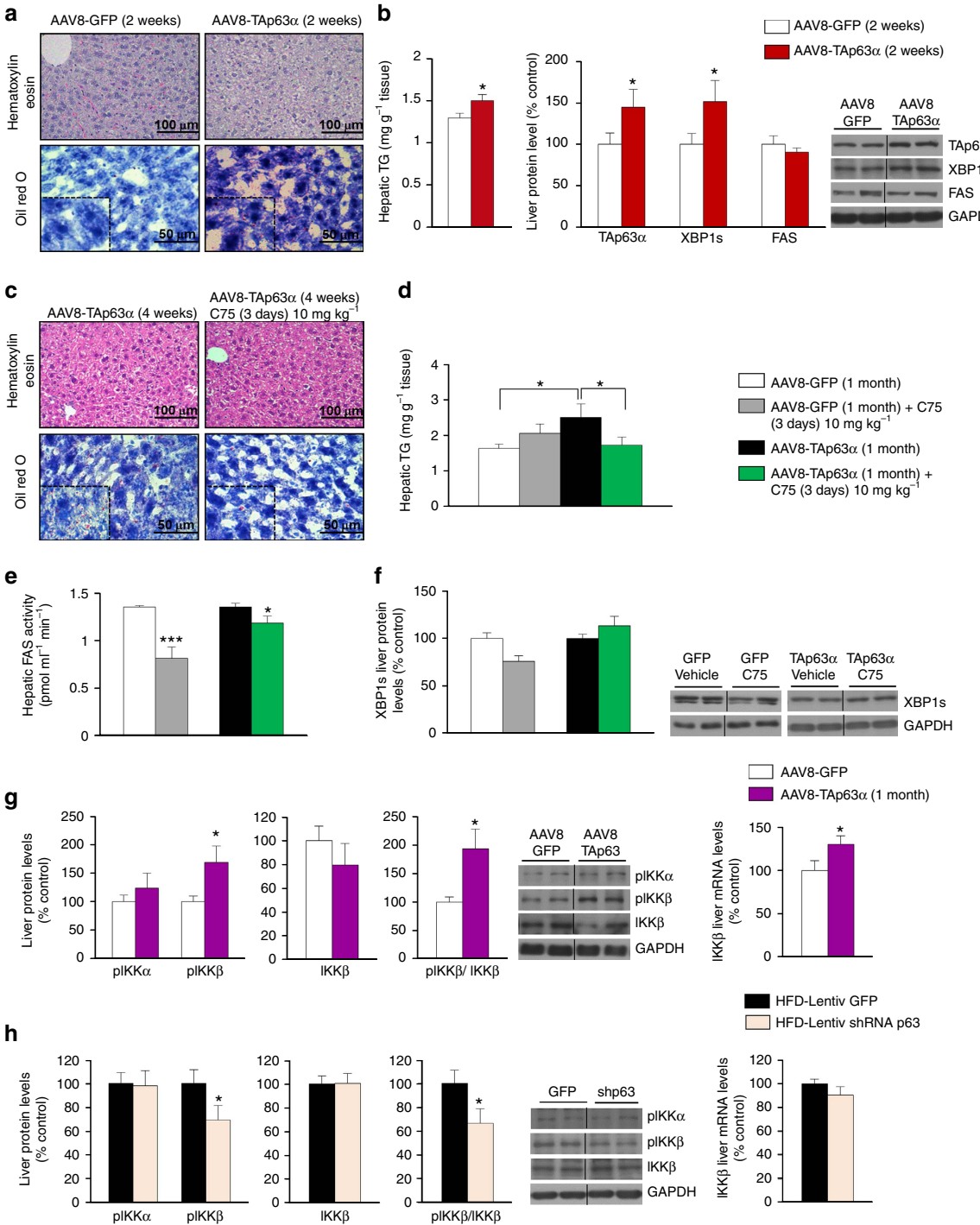

**Figure 7 | TAp63α-induced ER stress precedes changes in FAS expression and IKKβ links p63 to ER stress.** (**a**) Representative photomicrographs of hematoxylin-eosin (upper panel) and oil red O staining (lower panel) of mice liver sections ($n = 3$ per group); (**b**) total liver TG content and protein levels of TAp63α, XBP1s and FAS in the liver of mice 2 weeks after hepatic over-expression of TAp63α ($n = 6$ per group). (**c**) Representative photomicrographs of hematoxylin-eosin (upper panel) and oil red O staining (lower panel) of mice liver sections ($n = 3$ per group); (**d**) total liver TG content ($n = 8$ per group); (**e**) hepatic FAS activity ($n = 8$ per group); and (**f**) protein levels of XBP1s in the liver of mice injected with AAV over-expressing TAp63α treated with the FAS inhibitor C75 ($n = 7$ per group). (**g**) pIKKα, pIKKβ, IKKβ protein and mRNA levels in the liver of mice after 2 or 4 weeks of the tail vein injection of an AAV8 over-expressing either GFP or TAp63α isoform ($n = 7$ per group). (**h**) pIKKα, pIKKβ, IKKβ protein and mRNA levels in the liver of mice after a tail vein injection of a lentiviral particle that encodes either GFP or a shRNA p63 ($n = 7$ per group). Protein GAPDH levels were used to normalize protein levels and control values (AAV8 GFP) were normalized to 100%. Dividing lines indicate splicings in the same gel ($n = 7$ per group). Uncropped blots of this Figure accompanied by the location of molecular weight markers are shown in Supplementary Fig. 17. Data are presented as mean ± standard error mean (s.e.m.). Statistical differences are denoted by *$P < 0.05$, **$P < 0.01$ and ***$P < 0.001$. For multiple comparison (**d**) a one way ANOVA followed by Bonferroni or Kruskal-Wallis test was performed. Student $t$-test was used in the other panels.

opposite interactions between p53 and p63 depending on the tissue or the status of the cells. For instance, p53-dependent apoptosis in response to DNA damage required p63 -and p73- in mouse developing brain and embryonic fibroblasts[29]. However, in a mouse model p63 -and p73- did not contribute to p53 tumour suppression function in lymphoma development[47]. More recent

studies have shown that p63 have p53-like and p53-independent tumour suppressor functions[48]. An important aspect of our study was to decipher the p63 isoform responsible for changes in hepatic lipid metabolism. Our findings indicate that TAp63α was the only isoform inversely regulated by p53 and positively regulated by HFD, while the other TAp63 and ΔNp63 isoforms

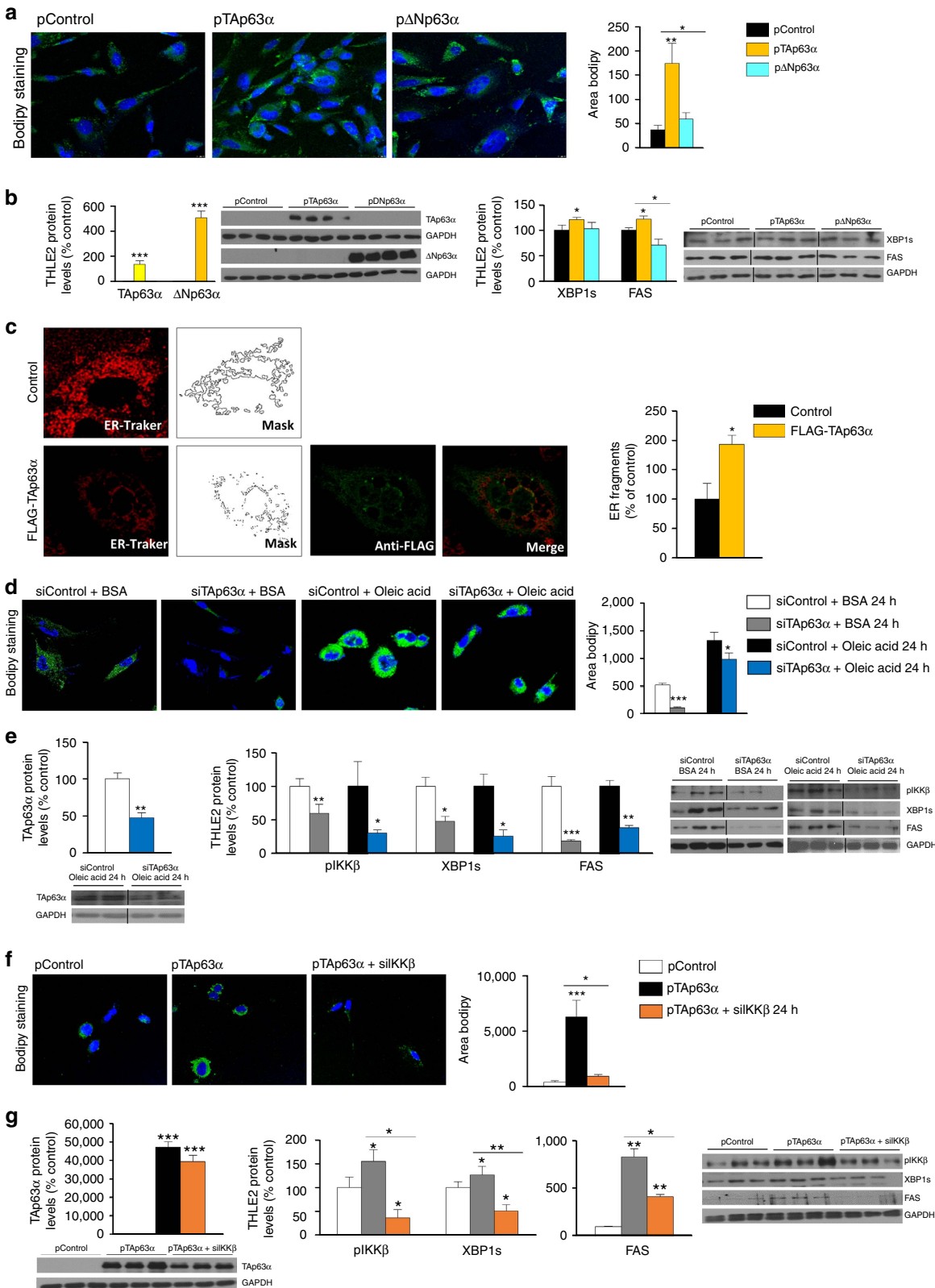

did not seem to be regulated by p53 or HFD. Consistent, our *in vitro* studies also suggest that over-expression of TAp63α but not ΔNp63α increases lipid content in THLE2 cells. Overall, our data indicate that TAp63α is the isoform controlling lipid metabolism in the liver. In this sense, it was reported that mice lacking the TAp63 in the whole body are obese and develop hepatic steatosis and insulin resistance[18]. Contrary to the obese phenotype in mice lacking TAp63 (ref. 49), heterozygous mice for total p53 showed weight loss[17], but the reason for this has not been investigated. The discordance between our results, indicating that TAp63 causes liver steatosis; and the previously report on mice lacking TAp63 suggest that the outcome of TAp63 deregulation is tissue-specific and that the global lack of TAp63 masks the role of TAp63 in the liver.

A key issue of the current study was to investigate the downstream molecular pathways controlling the hepatic actions of p63. The hepatic over-expression of p63 induces ER stress whereas its inhibition reduces ER stress. The importance of ER stress as a mediator of the hepatic functions of p63 is demonstrated by the fact that the pharmacological (TUDCA) or genetic (adenoviruses encoding GRP78) inhibition of ER stress blunts the steatotic action of TAp63α. The link between p63 and ER stress involves IKKβ as a transcriptional target, since IKKβ silencing was sufficient to blunt TAp63-induced lipid deposition and gene stimulation in hepatocytes. Although the association between IKKβ and ER stress has been found previously[50], earlier reports have shown different results regarding the association between p63 and IKK. For instance, one study investigating epidermal development showed that p63 acts upstream of IKKα, but IKKβ was not assessed[35]. To our knowledge, no studies have examined the possibility that other members of the IKK family can be modulated by p63. The fact that our results indicate that TAp63α increases phosphorylated levels of IKKβ is not surprising, if we take into account that IKKα has been shown to form a kinase complex with IKKβ and IKKγ, suggesting that any of those kinases might be modulated by p63. On the other hand, IKKβ, but not IKKα, was reported to inhibit TAp63γ (ref. 51), an unexpected observation inasmuch as IKKβ can also phosphorylate and stabilize TAp63γ (ref. 52). Given the complexity of the biology of p63, the available literature and our present results, we speculate that IKK and p63 can influence on each other reciprocally, likely dependent of the cell status. In this line, a similar reciprocal regulation has been reported for p53 and AMPK (see review[53]).

In any case, this is the first study indicating that TAp63α over-expression leads to IKKβ phosphorylation to cause an alteration of ER stress function that results in the activation of the UPR. Thus, IKKβ phosphorylation increases the levels of different markers for ER stress, and additionally, the chemical (TUDCA)

or genetic (over-expression of GPR78) amelioration of ER stress improved TAp63α-induced hepatic lipid storage. These results were also corroborated by microscopy indicating that the over-expression of TAp63α causes ER derangement and defragmentation[54]. The alteration of ER function ultimately leads to abnormal lipid storage in the liver. More precisely, the alteration of ER stress function caused by TAp63α lead to increased protein levels of FAS in the liver of different animal models and the augmented *de novo* TG lipogenesis in hepatocytes, without consistent changes in lipid oxidation or lipid turnover in the liver of mice or in THLE2 cells. Given the complexity of TAp63α actions, we cannot exclude that this transcription factor might be involved in different aspects of lipid metabolism in other cell types.

Finally, we investigated the potential clinical value of the current data and found that the inhibition of TAp63α in THLE2 cells protected against oleic acid-induced lipid accumulation, while the over-expression of TAp63α favoured the accumulation of lipid droplets. In line with these results, TAp63α -as well as IKKβ and Xbp1s- was increased in the liver from obese NAFLD/NASH patients, suggesting that the role of p63 in hepatic lipid metabolism may be important in human liver diseases. It should be pointed out that all our NAFLD/NASH patients were obese, and both NAS score and TAp63 showed a positive correlation with BMI, making it extremely difficult to dissociate BMI from liver damage. However, when we performed a Pearson correlation between TAp63 and NAS score, there was a positive correlation, suggesting that TAp63 is associated with NAFLD/NASH.

Overall, our findings indicate that gain- and loss-of-function experiments of p53 and p63 specifically in the liver of mice led to marked changes in hepatic lipid metabolism. TAp63α-induced liver steatosis is mediated by a IKKβ/ER stress/FAS pathway (Fig. 10f), and these results obtained in mice models were supported by data obtained in human hepatocytes and liver biopsies from obese NAFLD patients. Taken together, our data shed light on an unexpected role of the TAp63α/IKKβ/ER stress pathway in liver disease and identify p63 as a putative therapeutic target to tackle NAFLD.

## Methods

**Animals.** Animal experiments were conducted in accordance to the standards approved by the Faculty Animal Committee at the University of Santiago de Compostela, and the experiments were performed in agreement with the Rules of Laboratory Animal Care and International Law on Animal Experimentation. Wild type (WT) and p53-null (mixed background C57BL/6J and 129/Sv) mice were previously described[25] and were housed in air-conditioned rooms (22–24 °C) under a 12:12 h light/dark cycle. The reason for using a mixed background was that females are viable, whereas in a pure C57BL/6J background females are not viable. After weaning, mice were fed standard chow or high fat diet (Research Diets D12,451; 45% fat, 4.73 kcal g$^{-1}$ or D12,492; 60% fat, 5.24 kcal g$^{-1}$, Research Diets, New Brunswick, NJ) for 11 weeks.

**Figure 8 | Genetic manipulation of TAp63α regulates lipid content in THLE2 hepatocytes.** (**a**) Representative dual channel fluorescent photomicrograph of THLE2 cells showing staining of lipids for oleic acid (BODIPY 493/503, green) and nuclei (DAPI, blue) of THLE2 cells after the over-expression of TAp63α or ΔNp63α. Magnifications 63X. Right panel shows total lipid content (green area) ($n = 4$ per group). (**b**) Protein levels of TAp63α, ΔNp63α, XBP1s and FAS in THLE2 cells after the over-expression of TAp63α or ΔNp63α ($n = 4$ per group). (**c**) ER fragments in HepG2 cells transfected with empty plasmid control and plasmid encoding TAp63α. (**d**) Representative dual channel fluorescent photomicrograph of THLE2 cells showing staining of lipids for oleic acid (BODIPY 493/503, green) and nuclei (DAPI, blue) of THLE2 cells cultured in oleic acid medium, treated with empty siRNA control (left image) or siRNA TAp63α isoform (right image). Magnifications 63X. Right panel shows total lipid content (green area). (**e**) Protein levels of TAp63α, pIKKβ, XBP1s and FAS in THLE2 cells exposed to oleic acid, after treatment with siRNA against either control or TAp63α ($n = 3$ per group). (**f**) Representative dual channel fluorescent photomicrograph of THLE2 cells showing staining for oleic acid (BODIPY 493/503, green) and nuclei (DAPI, blue) of THLE2 cells transfected with empty plasmid control (left image), plasmid encoding TAp63α (middle image) and co-transfected with plasmid encoding TAp63α and siRNA IKKβ 24 h after transfection (right image). Magnifications 63X. Right panel shows total TG (green area). (**g**) Protein levels of TAp63α, pIKKβ, XBP1s and FAS in THLE2 cells after TAp63 over-expression followed by IKKβ silencing. GAPDH was used to normalize protein levels ($n = 3$ per group). Uncropped blots of this Figure accompanied by the location of molecular weight markers are shown in Supplementary Fig. 18. Data are presented as mean ± standard error mean (s.e.m.). Statistical differences are denoted by *$P < 0.05$, **$P < 0.01$ and ***$P < 0.001$. For multiple comparison (**a,b,f,g**) a one way ANOVA followed by Bonferroni or Kruskal-Wallis test was performed. Student $t$-test was used in the other panels.

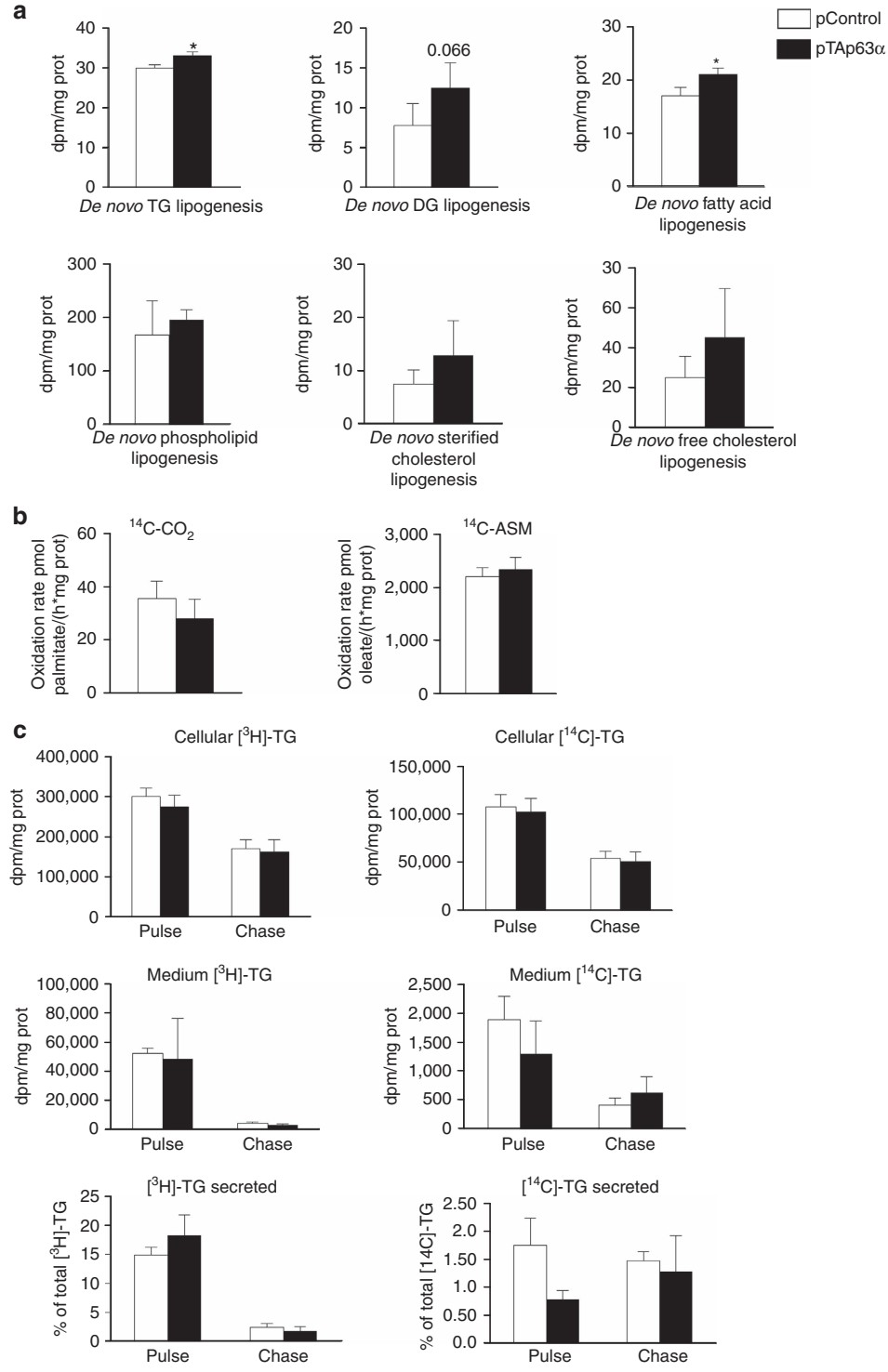

**Figure 9 | TAp63α stimulates lipogenesis in THLE2 cells. (a)** *De novo* triglyceride (TG), diacylglycerol (DG), fatty acid, phospholipid, esterified cholesterol and free cholesterol lipogenesis ($n = 4$ per group); **(b)** palmitate oxidation ($n = 4$ per group); and **(c)** lipid turnover in THLE2 cells transfected with empty plasmid control and plasmid encoding TAp63α ($n = 4$ per group). Data are presented as mean ± standard error mean (s.e.m.). Statistical differences are denoted by *$P < 0.05$, **$P < 0.01$ and ***$P < 0.001$, was tested using Student t-test.

Mice that carried floxed p53 alleles (with a C57BL/6 background) were obtained from The Jackson Laboratory. Mice used for the silencing or overexpression of p63 were males with a C57BL/6 background. Food intake and body weight were measured once every week during 11 weeks in high fat diet mice. Animals were killed by decapitation and tissues were removed rapidly and immediately frozen in dry ice. Tissues were kept at −80 °C until their analysis. Homozygous wild type and knockout mice were originated from heterozygous mating (Supplementary Fig. 1), so for each experiment only littermate WT and KO animals were compared.

**Determination of body composition.** Whole body composition was measured using NMR imaging (Whole Body Composition Analyzer; EchoMRI, Houston, TX).

**Hematoxylin/eosin staining.** Liver samples were fixed in 10% buffered formalin for 24 h, and then dehydrated and embedded in paraffin by a standard procedure. Sections of 3 μm were cut with a microtome and stained using a standard Hematoxylin/Eosin Alcoholic procedure according to the manufacturer's

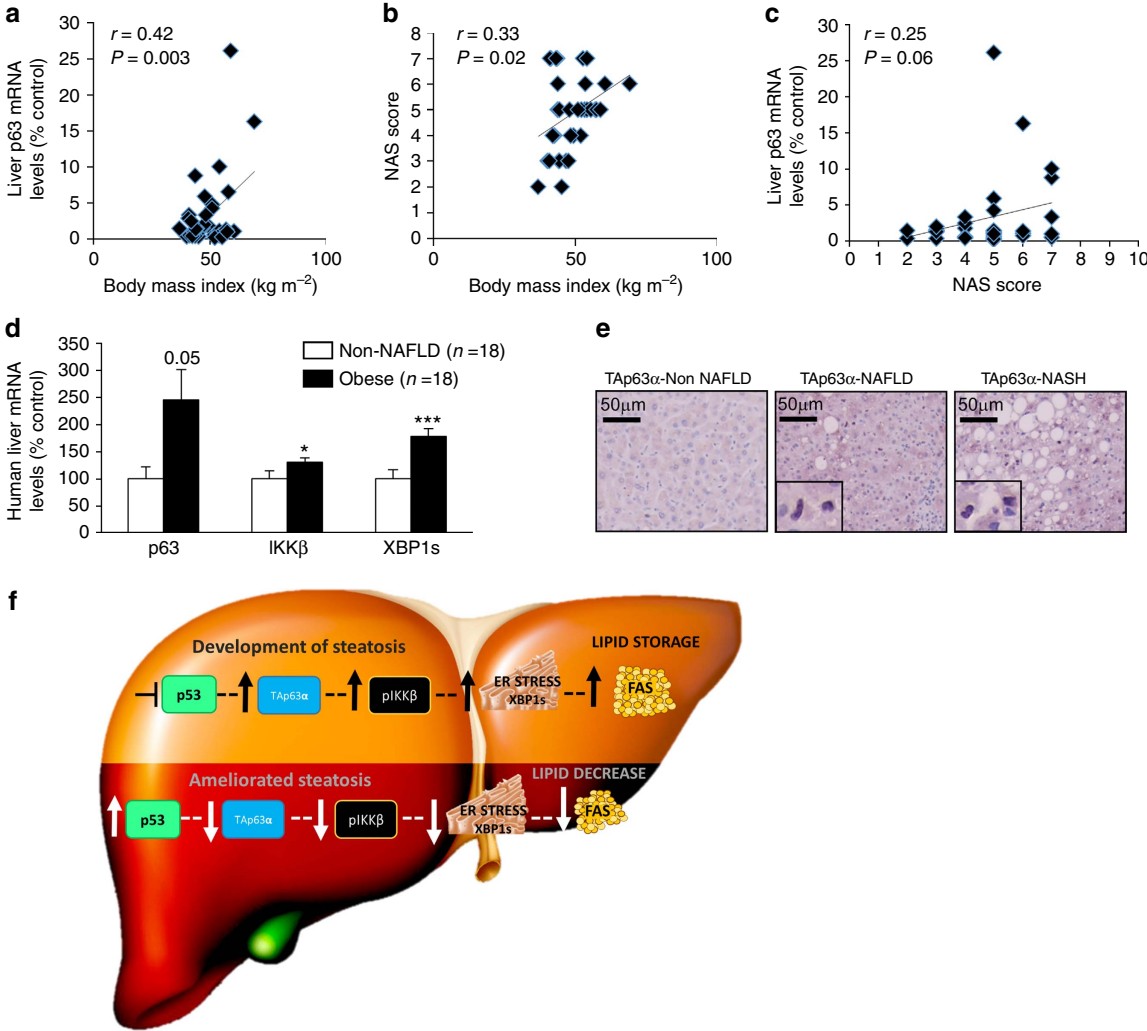

**Figure 10 | TAp63α is up-regulated in the liver of obese patients with NAFLD/NASH.** Correlations between (**a**) BMI and TAp63α; (**b**) BMI and NAS score; (**c**) TAp63α and NAS score; (**d**) Liver mRNA levels of TAp63, IKKβ and XBP1s in subjects without NAFLD ($n=18$) and with NAFLD ($n=48$). HRPT was used to normalize mRNA levels; and (**e**) Representative photomicrographs of an immunohistochemistry against TAp63 in human liver slices from lean ($n=11$), NAFLD ($n=23$) and NASH ($n=19$) patients. (**f**) Schematic representation of the pathway proposed to modulate lipid metabolism in liver. Data are presented as mean ± standard error mean (s.e.m.). Statistical differences are denoted by *$P<0.05$, **$P<0.01$ and ***$P<0.001$, was tested using Student $t$-test.

instructions (BioOptica, Milan, Italy). They were then mounted with permanent (non-alcohol, non-xylene based) mounting media, and evaluated and photographed using a BX51 microscope equipped with a DP70 digital camera (Olympus, Tokyo, Japan).

**Oil Red O staining.** Frozen sections of the livers (8 μm) were cut, fixed in 10% buffered formalin and stained in filtered Oil Red O for 10 min. Sections were washed in distilled water, counterstained with Mayers's haematoxylin for 3 min and mounted in aqueous mountant (glycerine jelly).

**TG content in liver.** Livers (approx 500 mg) were homogenized for 2 min in ice-cold chloroform-methanol (2:1, vol/vol). TG were extracted during 3-h shaking at room temperature. For phase separation, milli-Q water was added, samples were centrifuged, and the organic bottom layer was collected. The organic solvent was dried using a Speed Vac and re-dissolved in chloroform. TG content of each sample was measured in duplicate after evaporation of the organic solvent using an enzymatic method (Spinreact, Spain).

**Glucose and insulin tolerance tests.** Basal blood glucose levels were measured after an overnight fast (12 h) with an Accucheck glucometer (Roche). GTT and ITT were done after an intraperitoneal injection of either 2 g kg⁻¹ D-glucose (Sigma-Aldrich, USA) or 0.75 U kg⁻¹ insulin (Sigma-Aldrich, USA) and area under the curve values were determined.

**Levels of serum metabolites and hormones.** Mice were killed 4 h after the start of the light cycle. Whole trunk blood was collected and then spun for 15 min at 3,000*g* and 4 °C. Serum cholesterol, glucose, TG levels (Spinreact, Spain); FFA, ketone bodies levels (Wako, USA) and adiponectin (Millipore, USA) were assessed using commercial kits based on a colorimetric method. Serum insulin levels were measured by RIA following the manufacturer's instructions. Serum activities of ALT and AST were measured using the ALT and AST Reagent Kit (Biosystems, Spain) with a Benchmark Plus Microplate Spectrophotometer.

**Quantitative reverse transcriptase PCR (qRT-PCR) analysis.** RNA was extracted using Trizol reagent (ThermoFisher, USA) according to the manufacturer's instructions. Total RNA of 500 ng were used for each RT reaction, and cDNA synthesis was performed using the SuperScript First-Strand Synthesis System (ThermoFisher, USA) and random primers. Negative control reactions, containing all reagents except the sample were used to ensure specificity of the PCR amplification. For analysis of gene expression we performed real-time reverse-transcription polymerase chain reaction (RT-PCR) assays using a fluorescent temperature cycler (TaqMan; Applied Biosystems, USA) following the manufacturer's instructions and SYBR green reagent (Agilent Technologies, USA). The PCR cycling conditions included an initial denaturation at 50 °C for 10 min followed by 40 cycles at 95 °C for 15 s and 60 °C for 1 min. The oligonucleotide specific primers are shown in Supplementary Table 1. For analysis of the data, the input value of gene expression was standardized to the HPRT value for the sample group and expressed as a comparison with the average value for the control group. All samples were run in duplicate and the average values were calculated.

**Western blot analysis.** Total protein lysates from liver (20 μg) and THLE2 cells (2 μg) were subjected to SDS–PAGE, electrotransferred onto polyvinylidene difluoride membranes (Millipore) and probed with the indicated antibodies: Fatty Acid Synthase (FAS) (H-300) (sc-20,140), JNK 1/3 (C-17) (sc-474), XBP-1 (M-186) (sc-7,160), p-PERK (TH981) (sc-32,577), p63 (H-129) (sc-11,386), eIF2α (FL-315) (sc-11,386), pelF2α (Ser52) (sc-1,01,670), pIKKαβ (ser180/ser181) (sc-23,470-R), p21 (C19) (sc-397) (Santa Cruz Biotechnology, USA); Phospho-SAP/JNK (Thr183/Tyr185) (81E11) Rabbit mAb (#4,668), Caspase 3 (#9,665), Cleaved Caspase 3 (Asp175) (#9,664), Caspase 7 (#9,492), Cleaved Caspase 7 (Asp 198) (#9,491), p53 (1C12) (#2,524), Phospho-p53-Ser15 Antibody (#9,284), Bax Antibody (#2,772) (Cell Signaling, USA); Anti-IKKβ [EPR6,043] (ab124,957), Anti-IKKβ (#2,772) (Cell Signaling, USA); Anti-IKKβ [EPR6,043] (ab124,957), Anti-IKKβ (phosphor Y188) (ab1,94,519), Anti-p73 antibody (EP436Y) (ab40,658), Anti-IRE1 antibody (ab37,073), Anti-SHC (phosphor S36) [6E10] (ab54,518) (Abcam, UK); IRE 1 alpha [p ser 724] (NB100–2,323) (Novus Biologicals, USA); Anti-p63(TA) clone: Poly6,189 (cat:618,902), Anti-p63(ΔN) clone: Poly6,190 (cat:619,002) (Biolegend, USA); GFP (Living Colours , Clontech, Tokio, Japan). Monoclonal anti-GAPDH mouse (CB1,001) (Upstate, Lake Placid, NY). The conditions for each antibody are described in Supplementary Table 2. For protein detection we used horseradish peroxidase-conjugated secondary antibodies and chemiluminescence (Amersham Biosciences, Little Chalfont, UK). Protein levels were normalized to GAPDH for each sample. Uncropped blots accompanied by the location of molecular weight markers are shown in Supplementary Figs 10–18.

**Determination of apolipoprotein B in liver samples.** For the analysis of apolipoprotein B, liver protein lysates were subjected to SDS–polyacrylamide gel electrophoresis (SDS–PAGE) electrophoresis using 4–15% precast gels (BIO-RAD, USA). Proteins were transferred onto polyvinylidene difluoride membranes (Millipore, USA) and probed using a primary antibody for Anti-Apolipoprotein B (ab20,737, Abcam, UK) and transferrin I-20 (sc-22,597, Santa Cruz Biotechnology, USA) as loading control.

**Enzymatic assay of fatty acid synthase.** Assays were carried out in a 96-well microtitre plate. 100 μl of liver lysate (3 μg μl$^{-1}$ of protein) were incubated with a NADPH (200 μM)/acetil-CoA (100 μM)/dipotassium phosphate (33 mM)/monopotassium phosphate (67 mM) solution during 10 min at 37 °C. To start the reaction, 30 μl of malonil-CoA (600 μM)/dipotassium phosphate (33 mM)/monopotassium phosphate (67 mM) was added into well and FAS activity was measured spectrophotometrically by following the utilization of NADPH at 340 nm during 20 min.

**In vivo adenoviral gene transfer.** Wild type mice and p53-null mice were maintained on high fat diet during 11 weeks. After that, mice were held in a specific restrainer for intravenous injections: Tailveiner (TV-150, Bioseb, France). The injections into the veins were carried out using a 27 G × 3/8′′ (0.40 mm × 10 mm) syringe. Mice were injected with 100 μl of adenoviral vectors diluted in saline. Over-expression of hepatic p53 and p63 was achieved by tail vein injection of adenoviral vectors activating p53 (SignaGen Laboratories, USA, ref # SL100,777), p63 (SignaGen Laboratories, USA, ref # 189SL100,865) and GFP (SignaGen Laboratories, USA, ref # SL100,833) (1 × 10$^9$ VG ml$^{-1}$).

For the down-regulation of p53 specifically in the liver, we used associated adenoviruses serotype 8 (AAV8), a serotype that affords efficient and specific transgene expression in hepatocytes in vivo at low vector doses[55]. AAV8-Cre (1 × 10$^{10}$ VG ml$^{-1}$) (AAV8-GFP # SL100,833; AAV8-Cre-GFP #SL100,835 Signagen Laboratories, USA) were injected into p53$^{lox/lox}$ mice (8–10 weeks). Lentiviral sh RNA (1 × 10$^7$ TU ml$^{-1}$) (SMART choice Lentiviral Mouse Trp63 mCMV-turboGFP shRNA ref: SH01-040654-02-20, Thermo Fisher, USA) were injected in p53 floxed and obese mice.

Either dominant positive GRP78 adenovirus or GFP control (VQ Ad mGRP 78 WT 050,312 and Empty AflII/eGFP 101,311 respectively[33,56,57] (ViraQuest Inc., USA) were administered in a volume of 100 μl 10$^9$ PFU/0,2 ml in the tail vein of C57BL/6 fed during one week with a Methionine and Choline deficient diet (A02082002B) (Rerearch Diets, USA).

**C75 in vivo treatment.** Peripheral treatment with C75 (C5,490 Sigma Aldrich, USA) results in a decrease in FAS activity. After three weeks of AAV8p63 tail vein injection, C75 was administered intraperitoneally at a dose of 10 mg Kg$^{-1}$ twice during last week before killing the mice.

**Cell culture and adenoviral transduction.** THLE2 cells, a human liver cell line, was purchased from ATCC (The Global Bioresource Center). They were cultured in bronchial epithelial cell basal medium and additives (BEGM from Lonza/Clonetics Corporation, USA, BEGM Bullet Kit; CC3,170). 1.5 × 10$^5$ cells were seeded in a six-well plate for all experiments. Cells were tested for mycoplasma contamination. After 24 h in culture, cells were transfected with specific small-interference RNA (si-RNA) to knock down the expression of p63 total (C7-000110, Dharmacon, USA), TAp63 isoform (C7-000112, Dharmacon, USA) and IKKβ (L-003503-00-0005, Dharmacon, USA). Non-targeting siRNA was

use as negative control (Dharmacon, USA). Cells were transfected using Dharmafect 1 reagent form Dharmacon following the protocol: 10 μl of each siRNA (5 μM) diluted in 190 μl of optiMEM (Life Technologies, USA) mixed with 4.5 μl of Dharmafect1 diluted in 196.5 μl of optiMEM (Life Technologies, USA). This mixture was incubated in a final volume of 1.5 ml of complete medium supplemented with FBS 5% during 8 h. After that, the medium was replaced with fresh BEBM for another 8 h until cells were collected for protein and RNA extraction and immunofluorescence analysis. Total p63 and TAp63α isoform were silenced during 48 h while IKKβ was silenced for 24 h.

THLE2 cells were transfected with a DNA plasmid containing the sequence necessary to increase the expression of Tap63 alpha isoform (plasmid 27,008) (Addgene, USA), ΔNp63 isoform (plasmid 26,979) (Addgene, USA) and control pcDNA 3.1 ( + / − ). Lipofectamine 2,000 (Invitrogen) was used to transfect cells with following protocol: 4 μl of Lipofectamine 2000 diluted on 150 μl of optiMEM mixed with 2,5 μg of DNA diluted on 150 μl of optiMEM. This mixture was incubated in a final volume of 1,5 ml of complete medium supplemented with FBS 5% during 8 h. After that, the medium was replaced with fresh complete medium for another 8 h until cells were collected for protein and RNA extraction and immunofluorescence analysis.

**Oleic acid treatment in p63-silenced THLE2 cells.** After silencing total p63 and TAp63α in THLE2 cells, cell culture medium was supplemented with 1 mM of oleic acid (Sigma Aldrich, USA) bound to fatty-acid-free bovine serum albumin (BSA) (Capricorn) (2:1 molar ratio). Controls were supplemented with BSA alone. THLE2 cells were incubated with oleic acid and BSA during 12 h. After treatments, the coverslips were placed in a 24-well plate with 500 μl of Minimum Essential Medium Eagle (Sigma-Aldrich, USA) complete medium and incubated (37 °C) during 40 min with BODIPY (green) 493/503 (1:200 dilution) (Thero Fisher, USA). After incubation, coverslips were washed with phosphate-buffered saline (PBS) and fixed with 4% formaldehyde during 10 min. The coverslips were mounted in aqueous medium (FluoroGel #17985-10) with DAPI (D-9,542, Sigma-Aldrich, USA) (blue) (1:1,000) for preserving cell fluorescence. Confocal images were collected using a Leica confocal microscope (Leica A0B5-SPSX) equipped with a high grade colour corrected plan apochromat lens for confocal scanning × 63/1.32 objective. Leica Confocal Software was used for acquisition and analysis. Images are combinations of optical sections taken in the z axis at 0.5-μm intervals.

**De novo lipogenesis.** Cells were incubated overnight in Eagle's Minimum Essential Medium (EMEM) supplemented with 0.5% (w/v) fatty acid free bovine serum albumin. Then, cells were incubated for 4 h with fresh medium supplemented with 84 nM insulin and 20 μM acetate containing 20 μCi/ml [$^3$H]acetate (Perkin Elmer, USA)[58]. After the incubation, the cells were harvested, washed twice with ice cold phosphate buffered saline (PBS) (pH 7.4). Lipids were extracted and separated by thin layer chromatography. Lipid classes were visualized by exposure to iodine vapour, the corresponding bands were scraped, and the label incorporated into lipids was determined by scintillation counting and expressed relative to the cell protein.

**Turnover and secretion of triglycerides.** Cells were incubated for 6 h in EMEM with low glucose and 0.5% (w/v) fatty acid free bovine serum albumin. Cells were then incubated for 4 h with 0.4 mM [$^3$H]oleic acid (5 μCi per dish) and [$^{14}$C]glycerol (0.5 μCi per dish) (pulse). [$^3$H]oleic acid and [$^{14}$C]glycerol were from Amersham Radiochemicals (UK) and PerkinElmer Inc (USA), respectively. After a wash of 1 h, cells were incubated for 4 h (chase)[59]. After the pulse and chase, lipids from hepatocytes and incubation media were exhaustively extracted and separated by thin layer chromatography. Lipid classes were visualized by exposure to iodine vapour, the bands corresponding to triglycerides were scraped, and the associated [$^3$H] and [$^{14}$C] radioactivity was determined by scintillation counting. The percentage of the total [$^3$H]-TG and [$^{14}$C]-TG that was secreted to the media was calculated.

**Fatty acid oxidation.** The rate of fatty acid oxidation was determined by measuring the amount of $^{14}$CO$_2$ (complete oxidation) and the amount of $^{14}$C labelled acid-soluble metabolites (ASM) (incomplete oxidation) released. Cells were incubated overnight in low glucose medium. and were then incubated 4 h with 0.2 mM palmitate containing 0.5 μCi ml$^{-1}$ [$^{14}$C]-palmitate (Perkin Elmer Inc (USA) in low glucose medium. Medium was collected in a tube containing Whatman filter paper soaked with 0.1 M NaOH in the cap and 500 μl of 3 M perchloric acid were added to release the CO$_2$, which was captured in the filter paper. The acidified medium was centrifuged at 21,000g for 10 min to remove particulate matter. The radioactivity of CO$_2$ captured by the filter papers and the radioactivity in acid-soluble metabolites (the supernatants of the culture media) was measured by a scintillation counter.

**ER fragmentation in HepG2 cells.** HepG2 cell cultures transfected with a FLAG-TAp63 expression vector were stained with ER-Tracker Red and examined with TCS-SP2-AOBS confocal microscope (Leica Corp., Germany) fitted with a Plan-Fluar × 63 oil immersion objective (n.a. = 1.4) for visualization of ER

membranes. Cells expressing the FLAG-TAp63 construct were identified using an anti-FLAG antibody. Evaluation of the number and area of ER Tracker-positive objects per cell was assessed using ImageJ 1.36 (National Institutes of Health, USA) as follows: a binary threshold (40–255 grey levels) was applied to z projection images, and the number and area of immunoreactive objects displaying a size higher than 5 pixel was measured in each cell using the Analyse Particles function of this software.

**Human liver samples.** The study population included a group of obese adult patients with body mass index (BMI) ≥ 35 kg m$^{-2}$ and a liver biopsy compatible with NAFLD. Participants were recruited from patients undergoing elective bariatric surgery at the University Hospital of Salamanca. As controls, we included individuals with BMI < 35 kg m$^{-2}$ who underwent laparoscopic cholecystectomy for gallstones. Baseline characteristics of these groups are listed in Supplementary Table 3.

Participants were excluded if they had a history of alcohol use disorders or excessive alcohol consumption (> 30 g per day in men and > 20 g per day in women), chronic hepatitis C or B, or if laboratory and/or histopathological data showed causes of liver disease other than NAFLD. The study was approved by the Ethics Committee of the University Hospital of Salamanca and all subjects provided written informed consent to undergo liver biopsy under direct vision during surgery.

Data were collected on demographic information (age, sex and ethnicity), anthropomorphic measurements (BMI), smoking and alcohol history, coexisting medical conditions, and medication use. Before surgery, fasting venous blood samples were collected for determination of complete cell blood count, total bilirubin, aspartate aminotransferase (AST), alanine aminotransferase (ALT), total cholesterol, high-density lipoprotein, low-density lipoprotein, triglycerides, creatinine, glucose and albumin.

**TAp63 immunohistochemistry.** Well-characterized liver biopsies from NAFLD, NASH and control lean patients recruited at the Universitary Hospital Marqués de Valdecilla in Santander, Spain, were obtained after informed consent according to the principles embodied in the Declaration of Helsinki. The characteristics of these patients are described in Supplementary Table 4. Paraffin embedded liver samples were sectioned, dewaxed and hydrated. Immunohistochemistry was performed as previously described[60]. Afterwards, antigen was unmasked using citrate buffer pH 6.0 during 20 min at 97° (no boil in a PT link). After, samples were incubated with the purified anti p63 (TA) antibody from BioLegend at 1:100 at 4 °C overnight. On continuation, procedure was done according to standard protocols with EnVision + System HRP (Dako, Denmark). Finally, samples were incubated with Vector Vip substrate (Vectorlabs, USA) for colour development. Images were taken with a × 10 or × 20 objective from an AXIO Imager A1 microscope (Carl Zeiss AG, Germany). Quantification of staining intensity, average sum of intensities and stained area percentage of each sample were calculated using FRIDA software (FRamework for Image Dataset Analysis) http://bui3.win.ad.jhu.edu/frida/ .

**Data analysis and statistics.** Results are given as mean ± standard error mean (s.e.m.). Animals were excluded when an objective experimental failure was observed[61]. In the molecular analysis, values detected by the two folds of standard deviation observed were considered a failure in the technique and were discarded. Randomization was essential to avoid any systematic source of variation in liver triglyceride content. For that, all experimental groups were made with set of animals of the same sex, age and similar body weight. Studies were not blinded to investigators. The number of animals used in each study is listed in the figure legends. All experiments were performed once if not otherwise indicated in figure legends.

To test if the populations follows a Gaussian distribution, a normality test was performed (Kolgomorov–Smirnof test for $n$ between 5–7; Shapiro-Wilk test for $n$ ≥ 7). For normal distributions, parametric test was used; for two population comparisons, an unpaired $t$ tests (two-tailed for treatment and phenotyping experiment, one-tailed otherwise) were used as indicated in figure legends; for multiple comparison test, a one-way ANOVA followed by Bonferroni *post hoc* multiple comparison test, was performed[62]. For non-Gaussian distributions, non-parametric test was used; Man-Whitney test were used for two comparison test, and Kruskal-Wallis followed by Dunn *post hoc* test for multiple comparison. Data analysis was performed using GraphPad Prism Software Version 5.0a (GraphPad, San Diego, CA).

**Data availability.** All data generated or analysed during this study are included in this published article (and its Supplementary Information Files) or from the corresponding author upon reasonable request.

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

## Acknowledgements

We deeply thank Dr Manuel Serrano (Spanish National Cancer Research Center, CNIO, Spain) for kindly providing p53 null mice and critically reading the article. This work has been supported by grants from Ministerio de Economia y Competitividad (C.D.: BFU2014–55,871; R.N.: BFU2015-70,664-R; M.M.M.: BFU2013-44229-R; A.C.: SAF2016-79381-R, FEDER/UE; GS: SAF2013-43506-R; M.L.M.-C.: SAF2014-54658-R; M.L.: SAF2015-71026-R; P.A.: SAF2015-64352-R; B.G.-T.: FPI Severo Ochoa CNIC program SVP-2013-067639), Xunta de Galicia (M.L.: 2015-CP079; R.N.: 2015-CP080 and PIE13/00024), Comunidad de Madrid (G.S.: S2010/BMD-2326); Fondo de Investigaciones Sanitarias (M.M.: PI10/01692), Fundación SEEN (R.N.), GV-Departamento de Salud-2013111114 (to M.L.M.-C.), ISCIII: PIE14/00031 (to M.L.M.-C.), Junta Provincial de Bizkaia- AECC (to M.L.M.-C.), AECC (T.C.D.); Basque Department of Industry, Tourism and Trade (Etortek) (A.C.), the BBVA foundation (A.C.), Fundación AstraZeneca (R.N.) Centro de Investigación Biomédica en Red (CIBER) de Fisiopatología de la Obesidad y Nutrición (CIBERobn). CIBERobn is an initiative of the Instituto de Salud Carlos III (ISCIII) of Spain, which is supported by FEDER funds. The participation of A.C. and A.Z.-L. as part of CIBERONC was co-funded with FEDER funds. The research leading to these results has also received funding from the European Community's Seventh Framework Programme under the following grant: A.C.: ERC StG-336343; R.N.: ERC StG-281408 and G.S.: ERC StG-260464.

## Author contributions

B.P., M.F.F., T.C.D, C.I., M.I., P.I., J.C., A.Z.-L., J.F., B.G.-T., N.M., S.T., A.V., A.C., J.S.-C., C.P. and X.B. performed in vivo experiments and analytical methods. L.H.-C. and M.M. collected human samples. B.P., M.M.M., J.Z., C.D., G.S., M.L., P.A., M.L.M.-C. and R.N. conceived and designed the experiments, interpreted and discussed the data, reviewed and edited the manuscript. R.N. developed the hypothesis, secured funding, coordinated the project and wrote the manuscript.

## Additional information

DOI: 10.1038/ncomms16059 OPEN

# Corrigendum: Hepatic p63 regulates steatosis via IKKβ/ER stress

Begoña Porteiro, Marcos F. Fondevila, Teresa C. Delgado, Cristina Iglesias, Monica Imbernon, Paula Iruzubieta, Javier Crespo, Amaia Zabala-Letona, Johan Fernø, Bárbara González-Terán, Nuria Matesanz, Lourdes Hernández-Cosido, Miguel Marcos, Sulay Tovar, Anxo Vidal, Julia Sánchez-Ceinos, Maria M. Malagon, Celia Pombo, Juan Zalvide, Arkaitz Carracedo, Xabier Buque, Carlos Dieguez, Guadalupe Sabio, Miguel López, Patricia Aspichueta, María L. Martínez-Chantar & Ruben Nogueiras

*Nature Communications* 8:15111 doi: 10.1038/ncomms15111 (2017); Published online 8 May 2017; Updated 16 Jun 2017

The affiliation details for Paula Iruzubieta and Javier Crespo are incorrect in this Article. The correct affiliation details for these authors are given below:

Department of Gastroenterology and Hepatology, Marqués de Valdecilla University Hospital, Centro de Investigación Biomédica en Red de Enfermedades Hepáticas y Digestivas (CIBERehd). Infection, Immunity and Digestive Pathology Group, Research Institute Marqués de Valdecilla (IDIVAL), Santander, Spain.

