## [Peer review file · Nature Communications]

Reviewers' comments:

Reviewer #1 (Remarks to the Author):

Porteiro et al report a series of studies supporting the concept that p53 and p63 contribute to the regulation of lipid metabolism in the liver, based on tissue-specific regulation for p53 and p63 protein levels in vivo, and in a human-derived liver cell line (THLE2), and expression of p63 in liver tissue from non-obese vs. obese human subjects with NAFLD w/wo NASH. They also show that effects of p63 on liver lipid content are associated with increased levels of IKK β protein and activation of ER stress, which has previously been associated with changes in FAS levels and lipid accumulation.

Previous studies have examined effects of p53 on lipid metabolism, but the present study uses tissue-specific models to more selectively examine its role in the liver, and examine downstream mechanisms mediating this effect.

The relevance of these results to human disease are largely based on data related to TAp63 in liver from obese and non-obese subjects. But considerable additional information is needed to support these results and analysis.

Changes in FAS expression are associated with changes in liver lipid content. However, other mechanisms mediating effects on hepatic lipid content are not explored, including in cell culture studies where the accumulation of exogenous lipid (oleic acid) is measured.

The manuscript can further strengthened by addressing the following concerns.

Major concerns:

1. The studies associate changes in p53 and p63 with lipid content, and, in some studies, changes in levels of FAS. However, other aspects of lipid metabolism which also may contribute to changes in lipid content are not considered, including triacylglycerol (TAG) turnover, TAG secretion, and fatty acid oxidation. For example, Figure 6A shows that suppressing TAp63 expression increases lipid accumulation in THLE2 cells exposed to oleic acid, and Fig 6C shows that this is associated with reduced expression of FAS (Fig6C). However, it is not clear that changes in FAS synthesis (which promotes de novo lipogenesis) is involved in the accumulation of exogenous lipid in this study. Other aspects of lipid metabolism, including TAG turnover, TAG secretion, and fatty acid oxidation are more relevant to understanding how oleic acid is accumulated in these cells. This also is relevant since previous studies indicated that p53 (and presumably p63) exert effects on lipid metabolism at multiple levels, including both lipid anabolism and lipid catabolism, including fatty acid oxidation, as discussed in reference 10.

2. Additional information regarding the characterization of the human subjects studied in Fig 6 is needed. Baseline characteristics are said to be listed in "Appendix Table 1" on p. 9, but Appendix Table 1 was not included in the files for review. Related, Fig 6C shows that p63 mRNA levels are increased in morbidly obese (BMI > 35) subjects with NAFLD compared to subjects without morbid obesity (BMI of < 35) or NAFLD - what is the mean and range for BMI in the control group? How similar or different are the 2 groups. One wonders whether changes in p63 expression may represent differences in adiposity, not just the presence or

absence of NAFLD. Are p63 levels associated with liver fat content when adiposity is controlled for? How many patients had NAFLD alone vs. NASH? Related to this, is the positive Pearson correlation between TAp63 expression and NASH score limited to patients with NASH, or is the correlation largely based on values in patients with NAFLD, not NASH?

3. Additional controls would be helpful in several studies. For example in Fig 6, knocking down TAp63 alters the expression of ER stress-associated

3. Related. p. 11, last sentence. What does the phrase "the percentage of TAp63 positive samples" indicate? Presumably, all liver cells express some level of TAp63. Also, the correlations alluded to seem to be important, and presenting them in greater detail would be appropriate, including relations between Tap63 and each of the parameters contributing to the NASH score.

Minor comments:

4. p. 3, Abstract, line 6. The studies presented indicate that IKK β and ER stress are required for effects of TAp63 on hepatic fat content. This is not the same as mediating effects of TAp63, since other mechanisms may also contribute to effects on fat content (see comment 1).

5. p. 3, 3rd line from bottom. What does "lipotoxicity lipid storage" mean?

6. p. 5, 2nd paragraph, line 4. "As a matter of fact" is too strong a phrase. Previous studies indicate this, but the question remains an area for investigation.

7. p. 5, 3rd paragraph, line 2. Although technically correct, I found it confusing to use the term "steatosis" to refer to liver fat content in a general way. It is more commonly used to pathological states where hepatic lipid content - e.g, in NAFLD.

8. p. 6, first paragraph, subtitle. See comment 7 re the use of "steatosis".

9. p. 6, 2nd paragraph, last sentence. Fig 1E. Results for GTT studies are usually presented as glucose levels without normalization, and should be done so here.

10. p. 6, 2nd paragraph, last sentence. Fig 1 F. Results for ITT can be expressed relative to glucose levels at time 0, but then the y-axis should indicate this. But in that case, values are usually expressed as percent of baseline, and not in terms of concentration (mg/dL).

11. p. 7, first paragraph, last sentence. Note use of "steatosis".

12. p. 7, first paragraph, last sentence and Fig 2E. Please comment on the significance of changes in FAS, pJNK/JNK, ER stress markers and cleaved caspase 3. Why were these markers selected at this point in the studies, and what do the results signify?

13. p. 7, first paragraph. Other factors also may contribute to changes in lipid content, including the expression of glycolytic and lipogenic genes involved in promoting lipogenic metabolism, which may be suppressed, or genes involved in promoting lipolysis and/or fatty acid oxidation. Related - it would be helpful to know if circulating levels of triglycerides are altered, e.g. due to decreased secretion of VLDL, resulting in the accumulation of liver

triglyceride content.

14. p. 7, 2nd paragraph, first sentence. Please indicate where target gene of p53 have been implicated.

15. p. 7, 2nd paragraph. What is the relationship between p53 and p63? Is it expected that the expression of p63 would be increased when p53 is suppressed? Or is this surprising? This is important to comment on here, and possibly in the introduction and/or discussion.

16. related - is it surprising that p53 and p63 would have seemingly opposite effects on lipid metabolism? Please comment/discuss.

17. p. 8, 2nd line. Adenoviral vectors are widely used to target the liver, since they are selectively taken up by the liver. Is the same known to be the case for lentiviral vectors? If not, then it is important to demonstrate whether or not the effects of lentiviral injection are liver-specific.

18. p. 8, line 6. Why wait 1 month before looking at the effects of viral infection? Were shorter time points examined? Waiting this much time makes it more likely that these effects may be indirect.

19. p. 8, second paragraph. Same concerns regarding the use of lentivirus, one month delay before studies.

20. p. 10, paragraph 1. Line 9. Insert "were" before "...treated with the specific FAS..."

21. p. 10, first paragraph, last sentence. C75 inhibits FAS function. Is it known to also reduce levels of FAS protein? If not, why does this happen? Any explanation?

22. p. 10, 2nd paragraph, 2nd line. Insert "of" before IKK.

23. p. 10, 2nd paragraph, last line. Was the phosphorylation of IKK β measured? Where was this shown? If it was not measured, why is this indicated here? If it was not measured, then "indicating" is too strong.

24. p. 11, last paragraph. Lines 1-3. Did all patients with BMI > 35 have NAFLD? If not, does the expression of TAp63 distinguish between the presence and absence of NAFLD in subjects with BMI > 35? Put another way, is TAp63 increased in obese patients with NAFLD compared to obese patients without NAFLD? Or only in comparison to "lean patients" without NAFLD. Related - how "lean" is the "lean" group? Having a BMI < 35 is not considered lean.

25. p. 11, last paragraph, lines 4-5. Was immunohistochemical analysis for NAFLD and NASH quantified? If not, how was it determined that "immunohistochemical analysis corroborated that TAp63 is increased...."

Reviewer #2 (Remarks to the Author):

Review of Porteiro et al, Nat Comm

In this manuscript, the authors examine the roles of p53 and its orthologs, p63 and p73, in maintaining hepatic lipid homeostasis. The authors use genetic approaches to manipulate p53 and p63 specifically in the liver (p73 was not found to be altered by absence of p53 and so was not pursued). They found that, while loss of p53 leads to hepatic steatosis, it also led to upregulation of p63, and loss of both p53 and p63 was protective. Thus, the effects of loss of p53 can best be explained by the compensatory upregulation of p63. Throughout the manuscript, the authors include some readouts for ER stress, and make the claim that the effects of p63 are mediated through an axis involving IKK(beta), ER stress, and lipogenesis.

A strength of the paper is the use of genetic approaches to rigorously and combinatorially address the contributions of p53 and p63. I find the conclusion that p63 is responsible for the effects of p53 deletion to be warranted by the data. This seems to be a novel and interesting conclusion of this manuscript.

The proposed mechanistic linkage between p63 and lipid accumulation is much more problematic; it is clouded by the absence of some essential controls and also the use of data that are a bit superficial and in some cases not credible as presented. More specifically:

1. It cannot be concluded that p63 exerts its effects via ER stress for two reasons. Most importantly, the only functional testing of a role for ER stress is via administration of TUDCA. One possible reason this is problematic is that TUDCA is a bile salt, and the provision of bile salts is known to alter hepatic cholesterol homeostasis. At a minimum, the authors would need to carry out parallel experiments using PBA, which also acts as an apparent pharmacological chaperone in the liver. More significantly, though, is the fact that the TUDCA experiments are not controlled with animals not overexpressing p63 but treated with TUDCA (i.e., AAV8 GFP + TUDCA). To actually conclude that p63 exerts its effects through ER stress, the authors would need to show that TUDCA has no anti-steatotic effect in these animals (ideally in the same experiment as that using the p63-overexpressors with and without TUDCA). Otherwise, the simplest interpretation of the data as presented is that p63 induces steatosis and that TUDCA reduces steatosis, but that these effects are completely independent of each other.

2. The second reason that the conclusion is suspect is that the analysis of ER stress markers is superficial and in some cases not credible. I have little confidence that the bands shown are what they are purported to be. There are no molecular weight standards, the bands are tightly cropped, and there are no positive controls (e.g., overexpression of the indicated proteins) shown. As a concrete example, it is well known within the ER stress field that XBP1 protein can only be reliably detected following nuclear isolation, which was not (according to the methods) performed here. There is also no corroborating evidence of ER stress (mRNA expression of UPR target genes and conventional RT-PCR of Xbp1 mRNA would be particularly informative) to raise confidence in the data.

3. The problem of the missing essential control outlined in point #1 above also limits the conclusions that IKK mediates the effects of p63, and that this axis acts through lipogenesis. Particularly for the latter claim, one might expect that inhibiting FASN would reduce steatosis irrespective of whatever effects overexpressing p63 might have. Animals treated with or

without C75 but not overexpressing TAp63 are not shown, however. Likewise, animals not overexpressing TAp63 but expressing siIKK are not shown.

4. The authors examine expression of FASN throughout the manuscript, but really cannot come to any meaningful conclusion about what pathway or pathways of lipid metabolism are affected-not only because of the reason described above with respect to C75 treatment, but also because activity of lipogenic pathways are not examined. The effects of p63 could be due to effects on lipogenesis, FA oxidation, VLDL secretion, TG storage, uptake, etc.

Other points:

1. The authors state on p. 14 that "this is the first study indicating that TAp63 phosphorylates IKK(beta)..." This is not correct. TAp63 overexpression leads to IKK phosphorylation, but it cannot be concluded that p63 phosphorylates IKK.

2. I have mixed feelings about the fact that each experiment was conducted only once. While all of the experiments have an internal consistency that raises confidence in the data, the lack of replication is nonetheless worrisome. In addition, the authors describe criteria for exclusion of "outliers" based in part on their deviation from the means, but it is not clear how many animals this involves. In general, the exclusion of outliers for any reason other than a clear technical failure of the experiment in a given animal is problematic as it can lead to bias.

Reviewer #3 (Remarks to the Author):

Porteiro et al present an interesting study investigating the pathways downstream of liver-specific inactivation of p53, which is shown to induce steatosis, proposed to occur through a pathway involving p63 activation, IKKb phosphorylation and ER stress. The paper is clearly written and the data nicely presented. The story is of interest but could be strengthened by further investigation of endogenous p63 isoform expression, by reconciling the data with published literature, and by performing additional experiments to test the model as detailed below.

Specific Comments:

Introduction: Few if any studies have established that TAp63 is a tumor suppressor (but it may function as a metastasis suppressor), whereas DNp63 is established to have oncogenic properties (e.g. Keyes et al PMID 21295273, Ha et al PMID 21789189).

Fig. 1 It would be helpful throughout to indicate the timing of the experiments shown. How long after AAV injection do changes shown in B-D take to occur? How long do they persist?

Fig. 2. 2C-E. Are any differences between p53WT and p53KO significant in the absence of ectopic p53? It appears that the phenotype and expression changes are largely absent in WT vs. KO without acute activation of p53.

2F. The authors would definitely need to distinguish whether TAp63, DNp63 isoforms, or both are overexpressed in this context. Isoform-specific qRT-PCR and/or isoform-specific western blots would be required.

Secondly, can the authors show that changes in p63 levels precede the other alterations?

Finally, are any of the p53-associated liver alterations shown in Figs. 1-2 seen under the STD conditions? The differences in p63 levels in 2F are even greater than shown in 2H, so according to the model there should be recognizable p63-mediated liver changes.

Fig. 3. It is not clear the p63 expression shown in A is assessed under the same conditions as shown in B-D. Specifically, does the experiment shown in A include the AAV-Cre as well? If not, what does p63 expression look like under conditions shown in B-D where AAV-Cre is included?

3E-G. According to the model, one might expect the DIO protocol to lead to repressed p53 and consequently increased p63 compared to non-DIO mice. Are either of these observed?

Fig. 4. It is not clear why the authors chose TAp63 over DNp63 for these experiments. For example, in the epithelium the main relevant isoforms are DNp63. Secondly, which TAp63 C-terminal isoform is being ectopically expressed here? Activities of different C-terminal isoforms of TAp63 vary greatly, and it would strengthen the study if the authors could show that the particular isoform they are expressing is endogenously present in this context.

Fig. 5. Previous studies (refs 37, 38) showed that it is IKKa not IKKb that is downstream of p63, and other studies (e.g. Liao et al, PMID 23589370) showed that IKKb acts upstream of p63. Can the authors reconcile their findings with the published literature?

Fig. 6. Can the authors confirm that isoform-specific qRT-PCR and western blots are used to

detect TAp63 in 6G and H, respectively? Analysis of isoform-specific (TAp63 vs DNp63) expression of p63 in this context would be very helpful.

Minor comments:

List of abbreviations is incomplete, e.g. BAT, STD, HFD, DIO etc.

Reviewer #1 (Remarks to the Author):

The relevance of these results to human disease are largely based on data related to TAp63 in liver from obese and non-obese subjects. But considerable additional information is needed to support these results and analysis.

Changes in FAS expression are associated with changes in liver lipid content. However, other mechanisms mediating effects on hepatic lipid content are not explored, including in cell culture studies where the accumulation of exogenous lipid (oleic acid) is measured.

RESPONSE: We thank the Reviewer for the comments regarding the human relevance of our study. We agree that these experiments are important and would strength the conclusions of the paper. Therefore we have addressed all these comments by performing new experiments and further analysis of human data as detailed below. We would also like to point out that given the substantial amount of new data, the numbers of the figures have changed as indicated throughout the response to the Reviewers.

Major concerns:

1. The studies associate changes in p53 and p63 with lipid content, and, in some studies, changes in levels of FAS. However, other aspects of lipid metabolism which also may contribute to changes in lipid content are not considered, including triacylglycerol (TAG) turnover, TAG secretion, and fatty acid oxidation. For example, Figure 6A shows that suppressing TAp63 expression increases lipid accumulation in THLE2 cells exposed to oleic acid, and Fig 6C shows that this is associated with reduced expression of FAS (Fig6C). However, it is not clear that changes in FAS synthesis (which promotes *de novo* lipogenesis) is involved in the accumulation of exogenous lipid in this study. Other aspects of lipid metabolism, including TAG turnover, TAG secretion, and fatty acid oxidation are more relevant to understanding how oleic acid is accumulated in these cells. This also is relevant since previous studies indicated that p53 (and presumably p63) exert effects on lipid metabolism at multiple levels, including both lipid anabolism and lipid catabolism, including fatty acid oxidation, as discussed in reference 10.

RESPONSE: We totally agree with this comment, which was also made by another Reviewer. The assessment of not only lipogenesis, but also lipid oxidation and lipid turnover is indeed important for our study. Therefore, following the Reviewer's suggestion, we have done a substantial effort on this point.

- A. We measured lipogenesis in THLE2 cells (**new Figure 9A**). We found that after over-expression of TAp63 there was a significant increase in *de novo* TAG and fatty acid lipogenesis, and a non-significant tendency towards increased *de novo* DAG synthesis; while synthesis of *de novo* phospholipid, esterified cholesterol and free cholesterol biosynthesis remained unaltered.
- B. We measured lipid oxidation in THLE2 cells (**new Figure 9B**). We found that after the over-expression of TAp63 there were no differences in complete or incomplete palmitate oxidation rate in comparison to control cells.
- C. We measured lipid turnover in THLE2 cells by pulse and chase experiments using [³H]oleic acid and [¹⁴C]glycerol (**new Figure 9C**). We found that after the over-expression of TAp63 there were no changes in cellular [³H]-TG and [¹⁴C]-

TG, in the medium [³H]-TG, [¹⁴C]-TG or in the percentage of [³H]-TG, [¹⁴C]-TG secreted during pulse-chase studies.

D. We have also measured hepatic expression of genes involved in lipid oxidation and lipid turnover in the liver of several animal models, as follows:

- I. Mice with down-regulated p53 (**new Figure 1E**).
- II. p53-null mice and DIO mice with or without hepatic p53 (**new Figure 2E**).
- III. Mice with down-regulated p53 and p63 (**new Figure 5E**).
- IV. DIO mice with down-regulated p63 (**new Figure 5J**).
- V. Mice treated with TAp63 α + the chemical chaperone TUDCA (**new Figure 6E**).
- VI. Mice treated with TAp63 α + adenoviruses encoding the chaperone GRP78 (**new Figure 6J**).

These experiments failed to detect significant changes in the expression of genes involved in lipid oxidation such as CPT1, ACADM and ACADL or lipid uptake such as FATP2.

Overall, our results indicate that p63 specifically regulates *de novo* lipogenesis in the liver. Indeed, these results do not exclude the possibility that this transcription factor is involved in different aspects of lipid metabolism in other cell types. This is discussed on **pages 19-20**.

2. Additional information regarding the characterization of the human subjects studied in Fig 6 is needed. Baseline characteristics are said to be listed in "Appendix Table 1" on p. 9, but Appendix Table 1 was not included in the files for review. Related, Fig 6C shows that p63 mRNA levels are increased in morbidly obese (BMI > 35) subjects with NAFLD compared to subjects without morbid obesity (BMI < 35) or NAFLD - what is the mean and range for BMI in the control group? How similar or different are the 2 groups. One wonders whether changes in p63 expression may represent differences in adiposity, not just the presence or absence of NAFLD. Are p63 levels associated with liver fat content when adiposity is controlled for? How many patients had NAFLD alone vs. NASH? Related to this, is the positive Pearson correlation between TAp63 expression and NASH score limited to patients with NASH, or is the correlation largely based on values in patients with NAFLD, not NASH?

RESPONSE: We sincerely apologize for not including the Appendix Table in the files for review. We have now expanded the information of the human subjects studied. We obtained samples from two different cohort of patients, one used for measurement of TAp63 mRNA expression (**new Supplemental Table 3**) and another one used for TAp63 immunohistochemistry (**new Supplemental Table 4**). In both cases histological scoring was performed according to the NASH Clinical Research Network criteria (NASH CRN) (PMID: 15915461). Results representing p63 mRNA expression were shown in former Figure 6G (**new Figure 10D**). As indicated in the table, the mean and range for BMI in the control group is 27.2 \pm 4.24, while obese patients have a BMI 49.2 \pm 6.9. We have also performed correlations between BMI and p63 and these results

clearly show that there is a positive correlation (**new Figure 10A**). This is not surprising since patients with higher BMI also showed an increased NAS score (**new Figure 10B**), which makes it extremely difficult to dissociate BMI from liver damage. The point raised by the Reviewer is indeed interesting but due to the association of obesity with NAFLD and the ethical issues of performing a liver biopsy just for research purposes, it is extremely difficult to obtain liver samples from lean patients with NAFLD/NASH. However, when we performed a Pearson correlation between p63 and NAS score (including obese patients with NAFLD and NASH), there was a positive correlation (**new Figure 10C**), indicating that p63 is associated with NAFLD/NASH. Finally, the number of obese patients with NAFLD (n=8) and NASH (n=27) is now indicated in **new Supplemental Table 3**. Finally, we have added a new paragraph in the Discussion to address these issues.

3. Additional controls would be helpful in several studies. For example in Fig 6, knocking down TAp63 alters the expression of ER stress-associated

RESPONSE: We assume that the Reviewer is asking whether knocking down TAp63 α alters the expression of ER stress-associated proteins in cells not exposed to oleic acid. We have added these controls and our results indicate that knocking down TAp63 α in THLE2 cells under basal conditions also decreases the amount of lipids (**new Figure 8D**) and the protein levels of pIKK β , XBP1s and FAS (**new Figure 8E**). Thus, these results indicate that silencing TAp63 α is efficient under both basal and stimulated conditions.

3. Related. p. 11, last sentence. What does the phrase "the percentage of TAp63 positive samples" indicate? Presumably, all liver cells express some level of TAp63. Also, the correlations alluded to seem to be important, and presenting them in greater detail would be appropriate, including relations between Tap63 and each of the parameters contributing to the NASH score.

RESPONSE: We apologize for not writing this sentence more clearly. We meant that for immunohistochemistry (**new Figure 10E**) we have defined positive samples as those having at least n=5 nuclei of hepatocytes stained for p63. This has been now clarified in the text.

The table representing the characterization of NAFLD/NASH patients is now shown in **new Supplemental Table 4** and the corresponding correlations are now presented in detail in **new Supplemental Table 5**.

Minor comments:

4. p. 3, Abstract, line 6. The studies presented indicate that IKK β and ER stress are required for effects of TAp63 on hepatic fat content. This is not the same as mediating effects of TAp63, since other mechanisms may also contribute to effects on fat content (see comment 1).

RESPONSE: Agreed, the sentence has been modified.

5. p. 3, 3rd line from bottom. What does "lipotoxicity lipid storage" mean?

RESPONSE: We apologize if the sentence was not clear. We have now written: "oleic acid-induced lipid accumulation".

6. p. 5, 2nd paragraph, line 4. "As a matter of fact" is too strong a phrase. Previous studies indicate this, but the question remains an area for investigation.

RESPONSE: Agreed, the sentence has been modified.

7. p. 5, 3rd paragraph, line 2. Although technically correct, I found it confusing to use the term "steatosis" to refer to liver fat content in a general way. It is more commonly used to pathological states where hepatic lipid content - e.g., in NAFLD.

RESPONSE: Agreed, the term steatosis has been changed not only in line 2 but also in another 2 sentences of this paragraph (please see tracked changes).

8. p. 6, first paragraph, subtitle. See comment 7 re the use of "steatosis".

RESPONSE: Agreed, the subtitle has been modified.

9. p. 6, 2nd paragraph, last sentence. Fig 1E. Results for GTT studies are usually presented as glucose levels without normalization, and should be done so here.

RESPONSE: Agreed. Former Figure 1E (**new Figure 1G**) has been changed and the results remain the same, as we did not detect variations in glucose tolerance.

10. p. 6, 2nd paragraph, last sentence. Fig 1 F. Results for ITT can be expressed relative to glucose levels at time 0, but then the y-axis should indicate this. But in that case, values are usually expressed as percent of baseline, and not in terms of concentration (mg/dL).

RESPONSE: Apologies for this mistake. We have now corrected the values in the y-axis (**new Figure 1H**).

11. p. 7, first paragraph, last sentence. Note use of "steatosis".

RESPONSE: Agreed, the sentence has been modified.

12. p. 7, first paragraph, last sentence and Fig 2E. Please comment on the significance of changes in FAS, pJNK/JNK, ER stress markers and cleaved caspase 3. Why were these markers selected at this point in the studies, and what do the results signify?

RESPONSE: We agree with the Reviewer that clarifying the rationale for selecting these markers would help the readers to better understand the results. We have now explained the results in more detail, and taking into account that we have also measured markers for lipid oxidation and lipid uptake in the different animal models (**new Figures 1E, 2E, 5E, 5J, 6E and 6J**). We have also explained the rationale to measure these genes in the first paragraphs of the Results (**pages 6-7**). We believe that the scenario on lipid metabolism is much clearer now, since all the results point out to increased lipogenesis.

13. p. 7, first paragraph. Other factors also may contribute to changes in lipid content, including the expression of glycolytic and lipogenic genes involved in promoting lipogenic metabolism, which may be suppressed, or genes involved in promoting lipolysis and/or fatty acid oxidation. Related - it would be helpful to know if circulating levels of triglycerides are altered, e.g. due to decreased secretion of VLDL, resulting in the accumulation of liver triglyceride content

RESPONSE: We agree with the Reviewer that analysis of other markers is important for our study. As explained in point 1, we have now measured markers for lipid oxidation (CPT1, ACADM and ACADL) as well as a lipid uptake (FATP2) and secretion of VLDL (apoB) in the different animal models (**new Figures 1E, 2E, 5E, 5J, 6E and 6J**). These results, together with the assessment of lipogenesis, lipolysis and lipid turnover in cells (**new Figures 9A-9B-9C**) indicate that p63 specifically alters FAS expression and lipogenesis.

14. p. 7, 2nd paragraph, first sentence. Please indicate where target gene of p53 have been implicated.

RESPONSE: We now indicate that target genes of p53 have been implicated in hepatic lipid metabolism.

15. p. 7, 2nd paragraph. What is the relationship between p53 and p63? Is it expected that the expression of p63 would be increased when p53 is suppressed? Or is this surprising? This is important to comment on here, and possibly in the introduction and/or discussion.

RESPONSE: This is a very good question with a difficult answer. Different reports have shown opposite interactions between p53 and p63 depending on the tissue or the status of the cells. For instance, p53-dependent apoptosis in response to DNA damage

required p63 -and p73- in mouse developing brain and embryonic fibroblasts (Flores ER et al, *Nature* 2002). However, in a mouse model p63 -and p73- did not contribute to p53 tumor suppression function in lymphoma development (Perez-Losada J et al, *Oncogene* 2005). More recent studies have shown that p63 have p53-like and p53-independent tumor suppressor functions (Xu-Monette ZY et al, *Aging* 2016). In summary, the relationship between p53 and p63 can be quite different depending on the tissue and biological effect being assessed. In addition it should be noted the different effects of TAp63 and Δ Np63, raising the possibility that these isoforms may have similar/opposite or even unrelated actions to p53.

We have now clarified the reason for measuring p63 and p73 was that both p53 family member have been reported to mediate p53-dependent apoptosis. Moreover, we have expanded the discussion on this topic to clearly state the controversial relationship between p53 and p63 (**pages 17-18**).

16. related - is it surprising that p53 and p63 would have seemingly opposite effects on lipid metabolism? Please comment/discuss.

RESPONSE: As explained in the previous point, the relationship between p53 and p63 is very complex, but previous results obtained in the literature associating p53 and p63 in apoptosis prompted us to perform the measurement of p63. Furthermore, we were also aware of a study indicating that dehydrogenase/reductase member 3 (DHRS3), which facilitates lipid-droplet formation, is regulated by both p53 and p63 (Kirschner RD et al, *Cell Cycle* 2010). However, whether this interaction could be related to TAp63 or Δ Np63 was difficult to predict. Hence, we decided to assess in detail the expression of the different isoforms of p63 following gain-and-lost-of-function studies (**new Figure 3**) and also in mice fed a HFD (**new Figure 4**). When our results indicated that TAp63 α could be the most likely candidate mediating the effects of p53 on lipid metabolism, we carried out functional experiments assessing its effects. We have discussed these facts in the discussion (**pages 17-18**).

17. p. 8, 2nd line. Adenoviral vectors are widely used to target the liver, since they are selectively taken up by the liver. Is the same known to be the case for lentiviral vectors? If not, then it is important to demonstrate whether or not the effects of lentiviral injection are liver-specific.

RESPONSE: Lentiviral vectors are also used for the liver, as they preferably infect it when given in the tail vein (please see reviews: Manjunath N et al, *Adv Drug Deliv Rev.* 2009 or Pfeifer A et al, *Mol Ther.* 2001). There are also many examples of original paper using lentiviral delivery to target the liver (PMID: 27285989, PMID: 23149065, PMID: 23251393).

18. p. 8, line 6. Why wait 1 month before looking at the effects of viral infection? Were shorter time points examined? Waiting this much time makes it more likely that these effects may be indirect.

RESPONSE: The timing at which the viral infection is effective depends on the targeted gene and the end point that is to be measured. We waited 1 month to make sure that the effects caused by the viral infection were sustained over the time and not just a transient effect. We consider that this was important because NAFLD and NASH are chronic diseases and therefore we believe that results showing changes at long term are of higher physiological relevance and more attractive for the search of a new pharmacological gene target.

19. p. 8, second paragraph. Same concerns regarding the use of lentivirus, one month delay before studies.

RESPONSE: same answer as in previous point.

20. p. 10, paragraph 1. Line 9. Insert "were" before "...treated with the specific FAS..."

RESPONSE: Apologies for this mistake. We have now corrected this sentence.

21. p. 10, first paragraph, last sentence. C75 inhibits FAS function. Is it known to also reduce levels of FAS protein? If not, why does this happen? Any explanation?

RESPONSE: The injection of C75 has been shown to reduce FAS expression in previous reports (for example see Figure 6A in PMID: 12376313). Nevertheless, we agree that measurement of FAS activity is a more reliable way to assess FAS. Accordingly, we have replaced the graph showing FAS protein levels by a new one on FAS activity. As shown in **new Figure 7E**, the injection of C75 significantly decreases the activity of FAS.

22. p. 10, 2nd paragraph, 2nd line. Insert "of" before IKK.

RESPONSE: Apologies for this mistake. We have now corrected this sentence.

23. p. 10, 2nd paragraph, last line. Was the phosphorylation of IKK β measured? Where was this shown? If it was not measured, why is this indicated here? If it was not measured, then "indicating" is too strong.

RESPONSE: Phosphorylated levels of IKK β were measured by western blot (**new Figures 7G-7H**). We have modified the sentence accordingly.

24. p. 11, last paragraph. Lines 1-3. Did all patients with BMI > 35 have NAFLD? If not, does the expression of TAp63 distinguish between the presence and absence of NAFLD in subjects with BMI > 35? Put another way, is TAp63 increased in obese patients with NAFLD compared to obese patients without NAFLD? Or only in comparison to "lean patients" without NAFLD. Related - how "lean" is the "lean" group? Having a BMI < 35 is not considered lean.

RESPONSE: Yes, all patients with BMI > 35 have NAFLD (n=8) without NASH (NAS score < 4) or NASH (n=27). This is now indicated in **new Supplementary Table 3**. As explained above, we indicate all these details in the table. The mean and range for BMI in the control group is 27.2±4.24, while obese patients have a BMI 49.2±6.9. We have also performed correlations between BMI and p63 and these results show that there is indeed a positive correlation (**new Figure 10A**). This is not surprising since patients with higher BMI also showed an increased NAS score (**new Figure 10B**), being extremely difficult to dissociate BMI from liver damage. The point raised by the Reviewer is indeed interesting but due to the association of obesity with NAFLD and the ethic issues of performing a liver biopsy just for research purposes, it is extremely difficult to obtain liver samples from lean patients with NAFLD/NASH. However, when we performed a Pearson correlation between p63 and NAS score (including obese patients with NAFLD and NASH), there was a positive correlation (**new Figure 10C**), indicating that p63 is associated with NAFLD/NASH.

25. p. 11, last paragraph, lines 4-5. Was immunohistochemical analysis for NAFLD and NASH quantified? If not, how was it determined that "immunohistochemical analysis corroborated that TAp63 is increased...."

RESPONSE: For immunohistochemistry (**new Figure 10E**) we have assigned positive samples as having at least n=5 nuclei of hepatocytes stained for TAp63 α . This has been now clarified in the text. Also, details of the characterization of NAFLD/NASH patients is now shown in **new Supplemental Table 4** and the correlations alluded are now presented in detail in **new Supplemental Table 5**.

Reviewer #2 (Remarks to the Author):

A strength of the paper is the use of genetic approaches to rigorously and combinatorially address the contributions of p53 and p63. I find the conclusion that p63 is responsible for the effects of p53 deletion to be warranted by the data. This seems to be a novel and interesting conclusion of this manuscript.

The proposed mechanistic linkage between p63 and lipid accumulation is much more problematic; it is clouded by the absence of some essential controls and also the use of data that are a bit superficial and in some cases not credible as presented. More specifically:

RESPONSE: We thank the Reviewer for the comments regarding the relevance of the p53/p63 interaction. In addition, we also agree with the suggestions of the Reviewer to

improve the mechanistic aspect linking p63 with lipid accumulation. These experiments are important to strengthen the conclusions of the paper and therefore we have addressed all these comments by performing new experiments as detailed below. We would also like to point out that given the substantial amount of new data, the numbers of the figures have changed as indicated throughout the response to the Reviewers.

1. It cannot be concluded that p63 exerts its effects via ER stress for two reasons. Most importantly, the only functional testing of a role for ER stress is via administration of TUDCA. One possible reason this is problematic is that TUDCA is a bile salt, and the provision of bile salts is known to alter hepatic cholesterol homeostasis. At a minimum, the authors would need to carry out parallel experiments using PBA, which also acts as an apparent pharmacological chaperone in the liver. More significantly, though, is the fact that the TUDCA experiments are not controlled with animals not overexpressing p63 but treated with TUDCA (i.e., AAV8 GFP + TUDCA). To actually conclude that p63 exerts its effects through ER stress, the authors would need to show that TUDCA has no anti-steatotic effect in these animals (ideally in the same experiment as that using the p63-overexpressors with and without TUDCA). Otherwise, the simplest interpretation of the data as presented is that p63 induces steatosis and that TUDCA reduces steatosis, but that these effects are completely independent of each other.

RESPONSE: We totally agree with the Reviewer on this point and in order to clarify the relevance of ER stress in our study, we have now added the requested controls and also performed an additional experiment.

A. First, we injected TUDCA in mice not over-expressing p63. Unexpectedly, we found that the liver of these mice have increased levels of triglycerides (**new Figure 6B-6C**). We also measured FAS protein levels in the liver (**appendix Figure 1**) and detected higher levels of FAS in these normal mice treated with TUDCA. Although we do not have a clear explanation for this result, it might be related with the short treatment (1 week) or the fact that TUDCA is not able to decrease lipid deposition in the liver of healthy animals. In any case, it seems clear that in our experimental conditions TUDCA "*per se*" has no anti-steatotic effects in normal mice.

Appendix Figure 1. Representative protein levels of FAS in the liver of mice after hepatic over-expression of TAp63 alone or AAV8-TAp63+IP TUDCA. Protein GAPDH levels were used to normalize protein levels and control values (AAV8 GFP) were normalized to 100% (n=7 per group). Data are presented as mean \pm s.e.m. Statistical significance, *p<0.05, **p<0.01, and ***p<0.001.

B. We agree with the Reviewer that an additional experiment regarding p63 and ER stress could give further strength. Therefore we have over-expressed GRP78 (also named BIP), an ER chaperone that facilitates the proper protein folding acting upstream of the UPR (Schröder and Kaufman, 2005; Marciniak and Ron, 2006; Ron and Walter, 2007; Martinez de Morentin and Lopez, 2010; Fu et al., 2012; Cnop et al., 2012). Thus, an adenovirus encoding wild-type GRP78 (GRP78 WT) together with GFP or control adenovirus expressing GFP alone were injected in the tail vein (Imbernon M et al, Hepatology 2016). When both TAp63 and BIP were over-expressed in the liver, we found lower hepatic triglyceride levels and serum AST as well as a decrease in FAS, pIRE and XBP1s when compared to mice treated with AAV-Tap63 alone (**new Figures 6F-6G-6H-6I-6J**). Overall, these results support the hypothesis that GRP78, and thereby ER stress, acts downstream p63.

2. The second reason that the conclusion is suspect is that the analysis of ER stress markers is superficial and in some cases not credible. I have little confidence that the bands shown are what they are purported to be. There are no molecular weight standards, the bands are tightly cropped, and there are no positive controls (e.g., overexpression of the indicated proteins) shown. As a concrete example, it is well known within the ER stress field that XBP1 protein can only be reliably detected following nuclear isolation, which was not (according to the methods) performed here. There is also no corroborating evidence of ER stress (mRNA expression of UPR target genes and conventional RT-PCR of Xbp1 mRNA would be particularly informative) to raise confidence in the data.

RESPONSE: We apologize but with all due respect, we do not understand the “little confidence” of this Reviewer on the bands of our western blots. We prepared the figure showing only parts of the entire gels because otherwise it would be impossible to show all these proteins in each figure. As the Reviewer may be aware, this is the most common way to represent western blots by most research groups. For instance, we have followed the style of other papers published in Nature Communications (i.e. Serra H et al, *Nature Communications* 6, 7935, 2016; Martin-Martin N et al, *Nature Communications* 7, 12595, 2016) and we have also represented figures like this in previous articles and never had problems: Lopez M et al, *Nature Medicine* 2010; Imbernon M et al, *Gastroenterology* 2013; Martinez de Morentin P et al, *Cell Metabolism* 2014; Contreras C et al, *Cell Reports* 2014; Beiroa D et al, *Diabetes* 2014; Folgueira C et al *Diabetes* 2016; Imbernon M et al, *Hepatology* 2016, etc.

In order to solve any kind of doubt that the Reviewer has on our analysis of ER stress markers, we have done the following:

A. We present here (**appendix Figures 2 and 3**), the original films of western blots for the ER stress markers shown in **Figure 1D** and **Figure 6I** respectively.

Appendix Figure 2. Original films used for western blots of ER stress markers in Figure 1D of the manuscript.

Appendix Figure 3. Original films used for western blots of ER stress markers in Figure 6I of the manuscript.

- B. Following the recommendation of the Reviewer, we have also performed nuclear protein extraction and measured XBP1. Similarly to our results found on total protein levels, we found that: a) the deletion of hepatic p53 stimulates XBP1s following both nuclear and total protein isolation (**new Supplemental Figure 4**), and b) the over-expression of TAp63 α increases XBP1s following both nuclear and total protein isolation (**new Supplemental Figure 6**).

Overall, we hope that the new results and the photos provided to the Reviewer is now fully convinced.

3. The problem of the missing essential control outlined in point #1 above also limits the conclusions that IKK mediates the effects of p63, and that this axis acts through lipogenesis. Particularly for the latter claim, one might expect that inhibiting FASN would reduce steatosis irrespective of whatever effects overexpressing p63 might have. Animals treated with or without C75 but not overexpressing TAp63 are not shown, however. Likewise, animals not overexpressing TAp63 but expressing siIKK are not shown.

RESPONSE: We totally agree with the Reviewer on this point and in order to address it, we have now added the requested controls:

- A. Scramble or siRNAs directed against IKK β were transfected into THLE-2 cells under basal conditions (NOT challenged with oleic acid or NOT over-expressing TAp63 α). Lipid accumulation was unchanged after siRNA IKK β and accordingly, protein level of XBP1s and FAS were also unmodified (**new Supplementary Figure 7**).
- B. A group of normal (non AAV8-TAp63 α) mice was treated with C75 at the same dose and time than mice over-expressing TAp63 α . Under these conditions, we found that FAS activity was clearly inhibited (**new Figure 7E**), however, we were not able to find differences in hepatic triglycerides in comparison to the control group (**new Figure 7D**). Finally, protein levels of XBP1s were not affected by C75 (**new Figure 7F**), suggesting that changes in ER stress occur before changes in FAS.

4. The authors examine expression of FASN throughout the manuscript, but really cannot come to any meaningful conclusion about what pathway or pathways of lipid metabolism are affected-not only because of the reason described above with respect to C75 treatment, but also because activity of lipogenic pathways are not examined. The effects of p63 could be due to effects on lipogenesis, FA oxidation, VLDL secretion, TG storage, uptake, etc.**RESPONSE:** We totally agree with this comment, which was also made by another Reviewer. The assessment of not only lipogenesis, but also lipid oxidation and lipid turnover is indeed important for our study. Therefore, we have invested a substantial effort on this point.

- A. We measured lipogenesis in THLE2 cells (**new Figure 9A**). We found that after over-expression of TAp63 there was a significant increase in *de novo*

TAG and fatty acid lipogenesis, a non-significant tendency towards increased *de novo* DAG synthesis, while synthesis of *de novo* phospholipid, esterified cholesterol and free cholesterol biosynthesis remained unaltered.

- B. We measured lipid oxidation in THLE2 cells (**new Figure 9B**). We found that after the over-expression of TAp63 there were no differences in complete or incomplete palmitate oxidation rate in comparison to control cells.
- C. We measured lipid turnover in THLE2 cells by pulse and chase experiments using [³H]oleic acid and [¹⁴C]glycerol (**new Figure 9C**). We found that after the over-expression of TAp63 there were no changes in cellular [³H]-TG and [¹⁴C]-TG, in the medium [³H]-TG, [¹⁴C]-TG or in the percentage of [³H]-TG, [¹⁴C]-TG secreted during pulse-chase studies.
- D. We have also measured hepatic expression of genes involved in lipid oxidation and lipid turnover in several animal models:
 - I. Mice with down-regulated p53 (**new Figure 1E**).
 - II. p53-null mice and DIO mice with or without hepatic p53 (**new Figure 2E**).
 - III. Mice with down-regulated p53 and p63 (**new Figure 5E**).
 - IV. DIO mice with down-regulated p63 (**new Figure 5J**).
 - V. Mice treated with TAp63 α + the chemical chaperone TUDCA (**new Figure 6E**).
 - VI. Mice treated with TAp63 α + adenoviruses encoding the chaperone GRP78 (**new Figure 6J**).

These experiments failed to detect consistent changes in the expression of genes involved in lipid oxidation such as CPT1, ACADM and ACADL or lipid uptake such as FATP2.

Overall, our results indicate that p63 specifically regulates lipogenesis in the liver. Indeed, these results do not exclude the possibility that this transcription factor is involved in different aspects of lipid metabolism in other cell types. This is discussed on **pages 19-20**.

Other points:

1. The authors state on p. 14 that "this is the first study indicating that TAp63 phosphorylates IKK(beta)..." This is not correct. TAp63 overexpression leads to IKK phosphorylation, but it cannot be concluded that p63 phosphorylates IKK.

RESPONSE: Apologies for this mistake. We have now corrected this sentence.

2. I have mixed feelings about the fact that each experiment was conducted only once. While all of the experiments have an internal consistency that raises confidence in the data, the lack of replication is nonetheless worrisome. In addition, the authors describe criteria for exclusion of "outliers" based in part on their deviation from the means, but it is not clear how many animals this involves. In general, the exclusion of outliers for any reason other than a clear technical failure of the experiment in a given animal is problematic as it can lead to bias.

RESPONSE: We share with the Reviewer the need of replication of the most relevant findings in any of our papers. In this context, please note that many of the comparisons have been repeated in different experiments; i.e. knock-down of hepatic p53 was done in two set of p53-floxed mice, p53 over-expression was done in p53 KO mice and in mice fed a HFD, overexpression of TAp63 α was done in the experiments with TUDCA and BIP, and both gain and lack of function experiments for TAp63 were done not only in mice but also in cells. In summary, we respectfully believe that many experimental paradigms were done in different experiments and the results were also consistent using different set of mice.

Regarding the exclusion of "outliers" and the criteria used, we would like to emphasize that we wanted to clearly state that this was done correctly. However, we agree that this part of the paragraph could be better explained. The animals were excluded when an objective experimental failure was observed, like state of health, and those were not used for molecular analysis (Landis SC, Amara SG, Asadullah K, Austin CP, Blumenstein R, Bradley EW, Crystal RG, et al. A call for transparent reporting to optimize the predictive value of preclinical research. *Nature* 2012;490:187-191.). For analytical measurements, the values detected by the 2 folds of standard deviation observed were considered a failure in the molecular techniques (former reference 54). This later requirement did not apply for animal data, like body weight and food intake. The number of animals used in each study and technique is listed in the figure legends. The paragraph "*Data Analysis and Statistics*" has been modified accordingly. We hope that this aspect is now fully clarified.

Reviewer #3 (Remarks to the Author):

Porteiro et al present an interesting study investigating the pathways downstream of liver-specific inactivation of p53, which is shown to induce steatosis, proposed to occur through a pathway involving p63 activation, IKK β phosphorylation and ER stress. The paper is clearly written and the data nicely presented. The story is of interest but could be strengthened by further investigation of endogenous p63 isoform expression, by reconciling the data with published literature, and by performing additional experiments to test the model as detailed below.

RESPONSE: We thank the Reviewer for the comments regarding the interest of our results. We also agree with the suggestions of the Reviewer on p63 isoforms, as these experiments would improve the quality of the paper. Therefore, we have addressed all these comments by performing new experiments as detailed below.

Specific Comments:

Introduction: Few if any studies have established that TAp63 is a tumor suppressor (but it may function as a metastasis suppressor), whereas DNp63 is established to have oncogenic properties (e.g. Keyes et al PMID 21295273, Ha et al PMID 21789189).

RESPONSE: We apologize for not being very precise on the biology of p63. Our aim was to simplify the introduction to make it understandable for a broad audience. Moreover, we wrote tumor suppressor in accordance with other reports indicating that (i.e. PMID: 15837625, PMID: 19898465) TAp63 may be considered as a tumor suppressor. Anyhow, we have now modified this paragraph and clarified the distinct roles of TAp63 and Δ Np63.

Fig. 1 It would be helpful throughout to indicate the timing of the experiments shown. How long after AAV injection do changes shown in B-D take to occur? How long do they persist?

RESPONSE: We agree with the Reviewer that this information is important. We now state in the text that these effects were detected after 1 month of the injection of AAVs. We did not perform a time-response experiment in this study, but we are now extending our findings regarding the knock-down of hepatic p53 and nutrient availability, and appears to be clear changes after 2 months of the injection of AAVs (data not shown in the present manuscript). Thus, we speculate that AAVs, which are known to be active for months, can down-regulate the expression of p53 for a very long-term.

Fig. 2. 2C-E. Are any differences between p53WT and p53KO significant in the absence of ectopic p53? It appears that the phenotype and expression changes are largely absent in WT vs. KO without acute activation of p53.

RESPONSE: The differences between p53WT and p53KO mice were shown in detail in **Supplementary Figure 1** (body weight, body composition and glucose metabolism in males), **Supplementary Figure 2** (liver metabolism in males) and **Supplementary Figure 3** (body weight, body composition and glucose metabolism in females) and. Overall, mice lacking p53 show increased lipid storage in the liver when fed a chow or a high fat diet, and this is consistent with increased protein levels of FAS and ER stress markers (**Supplementary Figure 2**).

Regarding Figure 2, (**new Figure 2D**), we did not indicate the statistical differences between p53WT and p53KO in order to simplify the figure. Since the differences between WT and p53KO are shown in Supplementary Figures, the aim of **new Figure 2**

is to show just the effects of the over-expression and recovery of hepatic p53 in diet-induced obese mice and global p53-null mice respectively. Regarding protein levels (**new Figure 2F**), we performed the blots separately in WT and p53 KO, since the aim of this figure was to investigate the effect of hepatic p53 as explained before, and consequently these blots cannot be compared.

Additional data were included in this figure as requested by other Reviewers. More precisely, markers of lipid oxidation and uptake (**new Figure 2E**) and hepatic and circulating levels of apoB (**new Figure 2F**) were measured.

2F. The authors would definitely need to distinguish whether TAp63, DNp63 isoforms, or both are overexpressed in this context. Isoform-specific qRT-PCR and/or isoform-specific western blots would be required. Secondly, can the authors show that changes in p63 levels precede the other alterations?

RESPONSE: We agree with the Reviewer that given the complexity of p63, it is important to show which isoform is more relevant for our results. This comment prompted us to perform new measurements and one independent experiment, as described below. Given the increased amount of data, we present the results in **new Figures 3 and 4**, which are detailed as follows:

- A. We measured the expression of TAp63 and Δ Np63 isoforms in the liver of WT and p53-null mice fed a standard diet (**new Figure 3D**). The results indicate that TAp63 α is increased while TAp63 β and TAp63 γ remain unchanged. In this animal model, the levels of Δ Np63 β and Δ Np63 γ decrease, while Δ Np63 α does not change.
- B. We measured the expression of TAp63 and Δ Np63 isoforms in the liver of WT and p53-null mice fed a HFD with or without the ectopic expression of hepatic p53 (**new Figure 3E**). Our findings show that TAp63 α is decreased in both WT and p53-null mice where adp53+ was injected, thereby reproducing the results found in total p63 levels (**Figure 3B**). TAp63 β was decreased in WT mice treated with adp53+ but not in p53-null mice treated with adp53+; and TAp63 γ remain unchanged. In this animal model, the levels of Δ Np63 α and Δ Np63 β do not change, and Δ Np63 γ increased in WT mice treated with adp53+ but decreased in p53-null mice treated with adp53+.
- C. Since it is known that diet-induced obesity causes liver steatosis, we fed C57/B6 mice a standard diet (STD) and high fat diet (HFD) for 2 and 4 weeks, and then we measured by western blot TAp63 and Δ Np63 isoforms in the liver. We found that TAp63 α is the only transcript with a maintained high expression after 2 and 4 weeks on HFD, whereas TAp63 β is decreased and TAp63 γ is unchanged (**new Figure 4C**). When measuring Δ Np63 isoforms, the results were not very consistent as we detected higher levels of Δ Np63 α and Δ Np63 γ after 2 weeks of

HFD, whereas the levels of the 3 Δ Np63 isoforms decreased after 4 weeks of HFD (**new Figure 4C**).

Overall, these results indicate that TAp63 α is consistently altered by hepatic p53. More specifically, this isoform is increased in models of p53 deficiency and is decreased after the ectopic expression of p53 in the liver. These results, added to the new in vitro experiments comparing TAp63 α and Δ Np63 α (**new Figures 8A-8B**, described in detail below), indicate that TAp63 α is responsible for the changes in hepatic lipid metabolism.

Regarding the question whether changes in p63 levels precede the other alterations, **Figures 5-6-7-8** aim to demonstrate that manipulating p63 in the liver by gain- and loss-of-function experiments precedes changes in IKK β , ER stress and FAS.

Finally, are any of the p53-associated liver alterations shown in Figs.1-2 seen under the STD conditions? The differences in p63 levels in 2F are even greater than shown in 2H, so according to the model there should be recognizable p63-mediated liver changes.

RESPONSE: We apologize for not being clear with the type of diet in each experiment. **Figure 1** represents the results of the knock down of p53 in the liver of mice fed a standard diet. **Supplementary figure 2** represents data of WT and p53 KO mice fed a standard and high fat diet. Finally, **Figure 2** represents data of the rescue or over-expression of hepatic p53 in WT and p53KO mice fed a high fat diet. In summary, p53-associated liver alterations can be shown under both STD and HFD conditions.

Regarding former Figures 2F and 2H (**new Figures 3A and 3C**), they represent protein levels of total p63 in two different mice models; both of them fed a standard diet. In **Figure 3A** protein levels of p63 were measured in p53-null mice (mice used in Supplementary figure 3), whereas in **Figure 3C** protein levels of p63 were measured in mice after the knock down of p53 in the liver (mice used in Figure 1). The differences in p63 levels in Figure 3A are greater than shown in Figure 3C because in the first model there is a complete lack of p53, while in the second model the efficiency of the knock down in the liver was approx. 50% (please see Figure 1A).

Fig. 3. It is not clear the p63 expression shown in A is assessed under the same conditions as shown in B-D. Specifically, does the experiments shown in A include the AAV-Cre as well? If not, what does p63 expression look like under conditions shown in B-D where AAV-Cre is included?

RESPONSE: The Reviewer is right, former Figure 3A (**new Figure 5F**) was not assessed under the same conditions as former Figures 3B-D (**new Figures 5G-J**). We used the livers of mice fed a HFD that were injected with lentivirus shp63 as the lentivirus is exactly the same. Nevertheless, following the suggestion of the Reviewer, we have now added p63 levels for each condition (**new Figures 5A and 5F**).

3E-G. According to the model, one might expect the DIO protocol to lead to repressed p53 and consequently increased p63 compared to non-DIO mice. Are either of these observed?

RESPONSE: We thank the Reviewer for this excellent question. Regarding p53, it was previously described that its expression in the liver increases in different models of fatty liver disease (PMID: 14985341) and non-alcoholic steatohepatitis (PMID: 22641095). These results on p53 expression do not seem to fit with our model (also others, please see PMID: 24872453) that p53-deficiency in the liver leads to increased lipid accumulation. Our hypothesis is that the increased hepatic p53 expression observed in those animal models is not the cause of liver steatosis, but rather a consequence of the damage in the liver.

Regarding the expression of p63 in DIO mice, we performed a new experiment as explained above, feeding C57/B6 mice a standard diet (STD) and high fat diet (HFD) for 2 and 4 weeks, and then we used western blot to measure isoforms of TAp63 and Δ Np63. In that experiment, we found that TAp63 α is the only isoform with a maintained high expression after 2 and 4 weeks of HFD, whereas TAp63 β is decreased and TAp63 γ is unchanged (**new Figure 4C**). When measuring Δ Np63 isoforms, the results were not very consistent as we detected higher levels of Δ Np63 α and Δ Np63 γ after 2 weeks of HFD, whereas the levels of the 3 Δ Np63 isoforms decreased after 4 weeks of HFD (**new Figure 4C**).

Fig. 4. It is not clear why the authors chose TAp63 over Δ Np63 for these experiments. For example, in the epithelium the main relevant isoforms are Δ Np63. Secondly, which TAp63 C-terminal isoform is being ectopically expressed here? Activities of different C-terminal isoforms of TAp63 vary greatly, and it would strengthen the study if the authors could show that the particular isoform they are expressing is endogenously present in this context.

RESPONSE: Again we thank the Reviewer for this very interesting comment. We have now addressed more precisely the importance of each TAp63 and Δ Np63 isoform. These new experiments include:

- A. As explained in previous points, the expression of TAp63 α is the only one maintaining a clear consistency when we knock-down p53 in the liver or when we feed mice a high fat diet (**new Figures 3 and 4**).
- B. We have performed additional experiments in cells and show that the over-expression of Δ Np63 α does not affect lipid content whereas TAp63 α increases lipid content (**new Figures 8A-8B**) and increases ER fragmentation (**new Figure 8C**).

For these reasons, we have chosen to manipulate TAp63 α .

Fig. 5. Previous studies (refs 37, 38) showed that it is IKK α not IKK β that is downstream of p63, and other studies (e.g. Liao et al, PMID 23589370) showed that IKK β acts upstream of p63. Can the authors reconcile their findings with the published literature?

RESPONSE: This is a very good point that, as the Reviewer noticed, we partially touched in the discussion. Although we did not show these data initially, in our experiments we also measured hepatic levels of pIKK α after the genetic manipulation of TAp63 α . However we failed to detect significant changes in pIKK α when TAp63 α was over-expressed in the liver (**new Figure 7G**) or when TAp63 α was knocked down in the liver (**new Figure 7H**).

As the Reviewer indicates, the study showing that p63 acts upstream of IKK α was focused on epidermal development, and apparently in that study IKK β was not assessed, so we cannot know if this kinase is also relevant. As a matter of fact, to our knowledge, no studies have examined the possibility that other members of the IKK family can be modulated by p63. The fact that our results indicate that TAp63 α increases phosphorylated levels of IKK β is not that surprising, in particular if we take into account that IKK α has been shown to form a kinase complex with IKK β and IKK γ , suggesting that any of those kinases might be modulated by p63.

On the other hand, as the Reviewer comments, IKK β , but not IKK α , was reported to inhibit TAp63 γ (PMID 23589370). However, this effect is really surprising since IKK β can phosphorylate and stabilize TAp63 γ (PMID: 18411264). In any case, it seems clear that IKK β is upstream of p63.

Given the complexity of the biology of p63, the available literature and our present results, we speculate that IKK and p63 can interact reciprocally, likely dependent of the cell status. In this sense, another example of conditional reciprocal regulation is the interaction between p53 and AMPK. Reduced nutrient or energy levels result in a failure to activate AMPK, which leads to the induction of p53 (see review: PMID: 19759539). However, p53 can directly inhibit cell growth through the direct activation of AMPK (PMID: 17409411), suggesting that p53 has a key role in coordinating the cessation of proliferation and growth under times of starvation (PMID: 15928081)(also reviewed in: PMID: 19759539).

We have now expanded the discussion on these results and we have also included these articles in our manuscript (**pages 19-20**).

Fig. 6. Can the authors confirm that isoform-specific qRT-PCR and western blots are used to detect TAp63 in 6G and H, respectively? Analysis of isoform-specific (TAp63 vs DNp63) expression of p63 in this context would be very helpful.

RESPONSE: As explained above, the isoforms measured in former Figure 6 (**new Figure 8**) were TAp63 α and Δ Np63 α . We have not distinguished total p63 (i.e. **Figures 3A-3B-3C**) or specific isoforms (i.e. **Figure 8**) throughout the entire manuscript.

Minor comments:

List of abbreviations is incomplete, e.g. BAT, STD, HFD, DIO etc.

RESPONSE: We apologize for not including all the abbreviations in the list. We have now amended it, and name them when they were cited only once in the text (i.e. BAT-brown adipose tissue).

Reviewers' comments:

Reviewer #1 (Remarks to the Author):

The authors have done additional studies and have addressed each of the concerns that I raised in my critique, and the manuscript has been strengthened as a result.

Reviewer #2 (Remarks to the Author):

The authors have responded to criticisms from reviewers with extensive revision of the manuscript. For the most part, these additions address the previous complaints, and the paper has thus been strengthened. I do not believe further revision is necessary.

As a side note:

Included among these revisions was the provision of raw western blot data, as one of my original objections was that the material as presented was not credible because the bands were tightly cropped, there were no molecular weight standards, and no controls (e.g., overexpression or knockout). Showing the raw data certainly enhances confidence in the western blots. However, the authors also suggest that showing western blots in this way is the standard practice in the field. To which I would reply that, indeed, that is standard practice in the field, and is one of the factors contributing to the crisis of reproducibility that modern biomedical science is facing. How for example, I might ask the authors as a devil's advocate, do they know that their ~125kd band produced by the pIRE1 antibody is actually pIRE1 and not some spurious band of roughly the right molecular weight? If it were my data, I would definitely want to convince myself first that it was actually IRE1 by including either knock down/knockout or overexpression controls. And, in fact, having looked at wild-type and Ire1-knockout cells before with various commercial pIRE1 antibodies, it is quite clear that simply being an ~125 kD band and reactive with a pIRE1 antibody is no guarantee that pIRE1 is being identified. And so on for other western blots. I'm not making here an argument that the authors need to further revise the paper. Rather, I am asking them to apply to their own data a more stringent standard of rigor than the relatively low bar that the field has set.

Reviewer #3 (Remarks to the Author):

Porteiro et al have provided further supportive data for their proposed model that liver-specific inactivation of p53 induces steatosis through a pathway involving p63 activation, an increase in phosphorylated IKKb, and ER stress. In particular, it is quite remarkable that most of the effects on lipogenesis and steatosis are attributed to activation of TAp63, since homozygous germline deletion of TAp63 has been shown to induce lipogenesis and steatosis (ref 23, and incorrectly referred to as reference 58 in the Discussion).

Specific Points:

Regarding original figure 2f and comment "Can the authors show that changes in p63 levels precede the other alterations?" The authors now provide data that overexpression of p63 can induce some of the lipid alterations, but they do not seem to address the original question, which was whether endogenous p63 activation temporally preceded the observed changes in lipid metabolism.

Figures 3-4 Showing transfected p63 isoforms side-by-side with the endogenous proteins (at least in representative blots) would provide more confidence that the bands shown on the western blots actually represent the protein species proposed. Some of the species shown are known to be notoriously difficult to detect (e.g. TAp63gamma) so I do not have high confidence that the bands shown actually corresponds to this protein.

It is interesting in Figure 7 that the authors show changes in levels of phosphorylated IKKb (pIKKb) caused by TAp63, but levels of mRNA and total protein are not shown, so it is not possible to conclude whether this represents a change in phosphorylation, protein stability, mRNA expression, or some combination of these. These are relatively straightforward analyses which might help lend some mechanistic weight to the paper.

Fig. 10 Are there correlation coefficients and p-values associated with panels A-C? Also it would be interesting to see whether the same correlations were observed with DNp63.

Minor:

Fig. 8B Presumably the bar colors are in error in the left graph?

Reviewer #2 (Remarks to the Author):

The authors have responded to criticisms from reviewers with extensive revision of the manuscript. For the most part, these additions address the previous complaints, and the paper has thus been strengthened. I do not believe further revision is necessary.

As a side note:

Included among these revisions was the provision of raw western blot data, as one of my original objections was that the material as presented was not credible because the bands were tightly cropped, there were no molecular weight standards, and no controls (e.g., overexpression or knockout). Showing the raw data certainly enhances confidence in the western blots. However, the authors also suggest that showing western blots in this way is the standard practice in the field. To which I would reply that, indeed, that is standard practice in the field, and is one of the factors contributing to the crisis of reproducibility that modern biomedical science is facing. How for example, I might ask the authors as a devil's advocate, do they know that their ~125kd band produced by the pIRE1 antibody is actually pIRE1 and not some spurious band of roughly the right molecular weight? If it were my data, I would definitely want to convince myself first that it was actually IRE1 by including either knock down/knockout or overexpression controls. And, in fact, having looked at wild-type and Ire1-knockout cells before with various commercial pIRE1 antibodies, it is quite clear that simply being an ~125 kD band and reactive with a pIRE1 antibody is no guarantee that pIRE1 is being identified. And so on for other western blots. I'm not making here an argument that the authors need to further revise the paper. Rather, I am asking them to apply to their own data a more stringent standard of rigor than the relatively low bar that the field has set.

REPLY: We appreciate this comment and also agree with the Reviewer about the crisis of reproducibility. Although further revision was not necessary, we took this comment very seriously and we have performed an additional experiment. We have used a control model to detect an increase in the levels of ER stress markers inducing liver steatosis by a methionine-choline-deficient diet and then having another group of mice with this diet and injected with a viral vector overexpressing GRP78 specifically in the liver (**Supplementary Figure 8**). In this experiment we found that MCD-induced levels of ER stress markers are ameliorated after the administration of a viral vector overexpressing GRP78 in the liver.

As the Reviewer may be aware, it is well established that the unfolded protein response (UPR) is activated in models of obesity and steatosis/steatohepatitis (for review see: Malhi H, Kaufman RJ. Endoplasmic reticulum stress in liver disease. *J Hepatol.* 2011. 54:795-809). For instance, we just published a paper (Contreras C, et al. Reduction of Hypothalamic ER Stress Activates Browning of White Fat and Ameliorates Obesity. *Diabetes.* 2016 Sep 15. pii: db151547. [Epub ahead of print]) reporting that HFD-induced ER stress is ameliorated by the overexpression GRP78. In addition, two years ago we also published that ceramides (also known to activate UPR) increase protein levels of these ER stress markers and this effects is ameliorated by the overexpression GRP78 (Contreras C et al, *Cell Reports* 2014). Finally, we also found that in different rat and mouse models, the protein levels of these ER stress markers are consistent with the augmented or reduced hepatic fat accumulation (Imbernon M et al, *Hepatology* 2016). In all

those papers we used the same antibodies as in the present study to detect changes in ER stress markers. Therefore, we are confident in our WBs since all the results provide a proof that the antibodies respond according to the abundant literature in the field.

We hope that our previous studies showing the expected changes in protein levels of ER stress markers, together with the new study demonstrating that the over-expression of GRP78 in the liver reduces protein levels of all the studied ER stress markers (**new Supplementary Figure 8**) are convincing and make the results obtained by western blot reliable.

Besides that, we have now added ALL the uncropped blots as Supplementary Figures (**Supplementary Figures 10-18**), so these images will be available to all the readers.

Reviewer #3 (Remarks to the Author):

Porteiro et al have provided further supportive data for their proposed model that liver-specific inactivation of p53 induces steatosis through a pathway involving p63 activation, an increase in phosphorylated IKKb, and ER stress. In particular, it is quite remarkable that most of the effects on lipogenesis and steatosis are attributed to activation of TAp63, since homozygous germline deletion of TAp63 has been shown to induce lipogenesis and steatosis (ref 23, and incorrectly referred to as reference 58 in the Discussion).

REPLY: We apologize for this mistake in the reference, which has been amended in the new version of the manuscript.

Specific Points:

Regarding original figure 2f and comment “Can the authors show that changes in p63 levels precede the other alterations?” The authors now provide data that overexpression of p63 can induce some of the lipid alterations, but they do not seem to address the original question, which was whether endogenous p63 activation temporally preceded the observed changes in lipid metabolism.

REPLY: We apologize for the misunderstanding. We have now measured protein levels of FAS and ER stress markers in the liver of mice fed a high fat diet during only 2 weeks. As previously shown in **Figure 4C**, we found that TAp63 α is significantly increased after only 2 weeks of high fat diet. Now, we also show that at this time point, FAS protein levels are decreased (as previously published: Shillabeer G et al, Hepatic and adipose tissue lipogenic enzyme mRNA levels are suppressed by high fat diets in the rat. *J Lipid Res.* 1990 Apr;31(4):623-31) and the levels of some markers of ER stress such as pIRE, XBP1s and pPERK are unmodified (**new Figure 4D**). These data demonstrate that endogenous TAp63 α up-regulation temporally precedes changes in lipid metabolism.

Figures 3-4 Showing transfected p63 isoforms side-by-side with the endogenous proteins (at least in representative blots) would provide more confidence that the bands shown on the western blots actually represent the protein species proposed. Some of the species shown are known to be notoriously difficult to detect (e.g. TAp63gamma) so I do not have high confidence that the bands shown actually corresponds to this protein.

REPLY: Blots for TAp63 isoforms other than TAp63 α were tricky, but after some changes in the protocol we were able to detect bands at the predicted molecular weights according to the data sheet provided by the company (**Appendix figure 1**). As the Reviewer will see in **Supplemental figure 13**, TAp63 α is easily detected with our antibody. If we perform longer time exposures it is also possible to detect bands at lower molecular weight corresponding to the molecular weights provided by the data sheet for TAp63 β and TAp63 γ .

Product Data Sheet

Purified anti-p63 (TA)

Catalog # / Size: 618901 / 50 μ l (5 Western blots)
618902 / 200 μ l (20 Western blots)

Clone: Poly6189

Isotype: Rabbit Polyclonal IgG

Immunogen: Modified peptide

Reactivity: Human, recognizes TA-isoforms

Preparation: The antibody was purified by antigen-affinity chromatography.

Formulation: This antibody is provided in phosphate-buffered solution, pH 7.2, containing 0.09% sodium azide and 50% glycerol.

Concentration: Lot-specific (please contact technical support for concentration and total μ g amount, or use our Lookup tool if you have a lot number.)

Storage: Upon receipt, store frozen at -20°C.

Applications:

Applications: WB - Quality tested
IF, IHC - Reported in the literature

Recommended Usage: Each lot of this antibody is quality control tested by Western blotting. Western blotting, suggested working dilution(s): Use 10 μ l per 5 ml antibody dilution buffer for each mini-gel. It is recommended that the reagent be titrated for optimal performance for each application.

Application References:

1. Ben Khalifa Y, et al. 2011. *PLoS Pathog.* 7:e1002256. (IF)
2. Zhou Y, et al. 2011. *Clin. Invest. Med.* 34:E184. (IHC)
3. Tordella L, et al. 2013. *PNAS.* 110:17969. PubMed
4. Curtis KM, et al. 2015. *PLoS One.* 10:123642. PubMed
5. Ahronian LG, et al. 2015. *PLoS One.* 10:123816. PubMed

RRID: AB_315895 (BioLegend Cat. No. 618901)
AB_2207171 (BioLegend Cat. No. 618902)

Total cell lysate from 293E cells (lane 1, 15 μ g) and 293E cells transfected with human p63 (TA) (lane 2, 15 μ g) were resolved by electrophoresis (4-20% Tris-Glycine gel), transferred to nitrocellulose, and probed with purified anti-p63 (TA) antibody (poly6189). Proteins were visualized using an HRP Donkey anti-rabbit IgG Antibody and chemiluminescence detection. Direct-Biot™ HRP anti- β -actin antibody (clone 2F1-1) was used as a loading control.

Description: p63 (TA) is a member of the p53 family. This protein has multiple isoforms TA p63- α , - β , - γ , and δ N p63- α , - β , - γ . The molecular weight of the TA p63 isoforms α =77 kD, β =62 kD, and γ =56 kD. This nuclear protein is essential for limb formation, epidermal morphogenesis, epithelial stem cell regeneration, δ N isoforms are antagonistic to p53; TA isoforms can transactivate p53 targets. The p63 δ N isoforms repress TA isoforms; mutant p53 abrogates p63 transactivation This protein interacts with p53, p73, and the various p63 TA- and δ N isoforms. The Poly6189 antibody recognizes human p63 TA isoforms and has been shown to be useful for Western blotting.

Other Names: p63, Isoforms TA p63- α , - β , - γ

Antigen References:

1. Yang A, et al. 1998. *Mol Cell.* 2:305.
2. Celli J, et al. 1999. *Cell.* 99:143.
3. Ghioni P, et al. 2002. *Mol Cell Biol.* 22:8659.
4. Waltermann A, et al. 2003. *Oncogene.* 22:5686.

Appendix Figure 1. Data sheet of the anti-TAp63 antibody used in this study. This antibody recognizes TAp63 α , TAp63 β and TAp63 γ . The corresponding molecular weights are underlined in red.

Nevertheless, we agree that to show appropriate controls is essential and therefore we have now done transfections of TAp63 gamma (plasmid available at Addgene). As we show now in **new Supplementary Figure 9**, cells transfected with TAp63 gamma show a clear band at the expected size when using the antibody described below. Therefore, we are confident that the bands correspond to this protein.

It is interesting in Figure 7 that the authors show changes in levels of phosphorylated IKKb (pIKKb) caused by TAp63, but levels of mRNA and total protein are not shown, so it is not possible to conclude whether this represents a change in phosphorylation, protein stability, mRNA expression, or some combination of these. These are relatively straightforward analyses which might help lend some mechanistic weight to the paper.

REPLY: We have now measured IKKb mRNA and total protein levels (**new Figures 7G and 7H**). While IKKb protein levels do not change after manipulation of TAp63 α , the results on mRNA expression showed that IKKb increases after the overexpression of TAp63 α but does not change when p63 is inhibited.

Fig. 10 Are there correlation coefficients and p-values associated with panels A-C? Also it would be interesting to see whether the same correlations were observed with DNp63.

REPLY: The correlation coefficients and p-values have now been added to **Figures 10A, 10B and 10C**.

In **Figure 8A** we have shown that the over-expression of DNp63 does not affect lipid content in human cells. Moreover, in **Figures 3E and 4C** we have not found any clear pattern in protein levels of DNp63 in rodent's liver. Taking into account: a) these findings on DNp63; and b) that it is extremely difficult to get human liver samples and therefore these samples are only used when strictly necessary; we do not see a clear hypothesis or robust data that justify the use of these human liver RNAs for measuring DNp63.

Minor:

Fig. 8B Presumably the bar colors are in error in the left graph?

REPLY: The Reviewer is right. We have now corrected this mistake.

REVIEWERS' COMMENTS:

Reviewer #3 (Remarks to the Author):

Porteiro et al have been responsive to the most recent critiques and have corrected the errors noted. They now provide further supportive data for their proposed model that liver-specific inactivation of p53 induces steatosis through a pathway involving p63 activation, an increase in phosphorylated IKKb, and ER stress. Overall the data support the general model proposed. I do not have specific suggestions for additional revisions.